# Pre-training Differentially Private Models with Limited Public Data

**Zhiqi Bu**[*]
Amazon

**Xinwei Zhang**[*†]
University of Southern California

**Sheng Zha**
Amazon

**Mingyi Hong**
University of Minnesota

**George Karypis**
Amazon

## Abstract

The superior performance of large foundation models relies on the use of massive amounts of high-quality data, which often contain sensitive, private and copyrighted material that requires formal protection. While differential privacy (DP) is a prominent method to gauge the degree of security provided to the models, its application is commonly limited to the model fine-tuning stage, due to the performance degradation when DP is applied during the pre-training stage. Consequently, DP is yet not capable of protecting a substantial portion of the data used during the initial pre-training process. In this work, we provide a theoretical understanding of the efficacy of DP training by analyzing the per-iteration loss improvement, through the lens of Hessian matrix for large neural networks. We make a key observation that DP optimizers' performance degradation can be significantly mitigated by the use of limited public data, which leads to a novel DP continual pre-training strategy. Empirically, using only 10% of public data and 90% of private data, our strategy can achieve DP accuracy of 41.5% on ImageNet-21k (with $\epsilon = 8$), as well as non-DP accuracy of 55.7% and 60.0% on downstream tasks Places365 and iNaturalist-2021, respectively, on par with state-of-the-art standard pre-training and substantially outperforming existing DP pre-trained models. Our DP pre-trained models are released in `fastDP` library (https://github.com/awslabs/fast-differential-privacy/releases/tag/v2.1).

## 1 Introduction

Large pre-trained models have been the backbone of computer vision and natural language processing. Finetuning or zero/few-shots learning (including in-context learning) based on these models can achieve superior performance. In particular, differentially private (DP) fine-tuning, such as full-parameter training, LoRA, Adapter, BiTFiT, and linear probing [89, 49, 14, 12, 58, 24], has shown to be almost as accurate as the standard non-DP fine-tuning on GPT [68, 6], ViT [27], and ResNet [39] models, while protecting the privacy of fine-tuning data. To be more specific, DP language and vision models are highly effective in defending against canary insertion attacks [66, 40, 42] and membership inference attacks [20, 69, 32, 16]; DP fine-tuned GPT2 also reduces the personally identifiable information leakage by $5 \sim 10$ times compared with its non-DP counterpart [53].

These DP fine-tuned models all follow a two-stage training procedure, in which the first stage trains a model on large public datasets (e.g. ImageNet) from scratch without privacy protection, and the second stage fine-tunes on relatively small private datasets (e.g. CIFAR10). However, a growing concern has been raised against the pre-training on the vast amount of web-collected data [18, 17, 79, 41, 62, 67]. The pre-trained models could memorize and re-generate the sensitive information in the

---

[*]Equal contribution. Email: zhiqibu@amazon.com. [†]Done at Amazon.

38th Conference on Neural Information Processing Systems (NeurIPS 2024).

training data, which include copyright content in books and codes, images of faces and nudity, and other personally identifiable information such as address and phone numbers, even if the data have been pre-processed by some content filters. While it is common to use close-source or proprietary datasets (e.g. JFT [75, 24, 58]) instead and not release the models trained on these datasets, this approach renders the models not reproducible and may still violate the data privacy implicitly.

Consequently, because of the uncertainty in seemingly safe-to-use public datasets, it is important to apply DP to the pre-training phase, which is computationally feasible, since DP optimization can be as efficient as the standard non-DP training [13, 38]. However, DP optimization without any public data suffers from slow convergence, and suboptimal performance: even on CIFAR10, the non-DP accuracy drops from $> 95\%$ to $< 70\%$ at $\epsilon = 8$ [63]; on GPT2, the non-DP BLEU score degrades from 65.73 (quality often better than human) to 15.457 (hard to get the gist) at $\epsilon = 3$ [49]. In short, pre-training and fine-tuning differ significantly in the amount of data and computation resources used, as well as in optimization setups. We summarize the difference in Table 1.

Table 1: Comparing DP pre-training and DP fine-tuning.

|  | pre-training | fine-tuning |
|---|---|---|
| Dataset size | large | small |
| Training iterations (amount of compute) | large | small |
| Trainable model parameters | 100% | $0.1 \sim 100\%$ |
| Major cause of performance degradation | DP noising | DP clipping |

## 1.1 Contributions

Our main contributions are listed below. We emphasize that we have taken the privacy accounting and the convergence (i.e. the dependence of loss on $\sigma(B, T, \epsilon)$ and $B$) jointly into consideration.

**(1)** We provide a new perspective of analyzing the loss improvement in DP training with different optimizers. Specifically, in Section 2, we propose a Hessian-based framework to analyze the per-iteration improvement on the generalization loss and the effects of per-sample gradient clipping, noising, and hyperparameter choices (e.g., batch size, learning rate).

**(2)** Based on the framework, we can analyze the effect that DP mechanisms, especially the DP noise but less so the DP clipping, have on pre-training and fine-tuning stages. This leads to some theoretical justifications about why pre-training is more vulnerable to DP noise; see Figure 1 for some empirical comparisons, and Section 3 for some in-depth discussion.

**(3)** Our analysis suggests that the deceleration due to DP mechanisms can be mitigated by using a certain amount of public data. We then propose a DP continual pre-training strategy that continues the pre-training privately from a non-DP initialization. Our strategy is easily implementable, automatic and effective, obtaining accuracy on upstream and downstream tasks that is on par with its non-DP variants, while significantly outperforming state-of-the-art DP pretraining algorithms (see Figure 2).

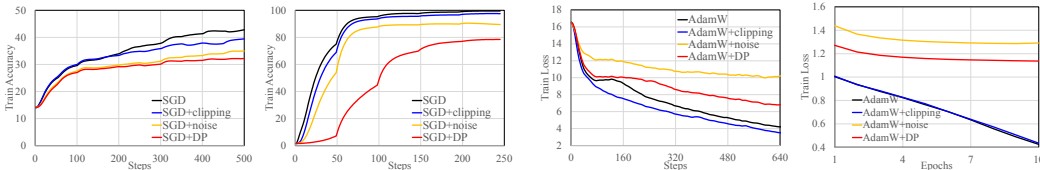

(a) Pre-training ViT-Base (CIFAR10,$\epsilon = 1$)  (b) Fine-tuning ViT-Large (CIFAR100, $\epsilon = 1$)  (c) Pre-training GPT2-Large (CodeParrot, $\epsilon = 8$)  (d) Fine-tuning GPT2-Large (E2E, $\epsilon = 1$)

Figure 1: Comparison among the convergence of standard SGD, clipped SGD without noise, noisy SGD without clipping, and DP-SGD in different tasks and training stages.

## 1.2 Related works

This work is closely related to other works in DP convergence analysis, DP fine-tuning of large models, the continual training and the Hessian-based analysis. We refer to Appendix C for an extended discussion.

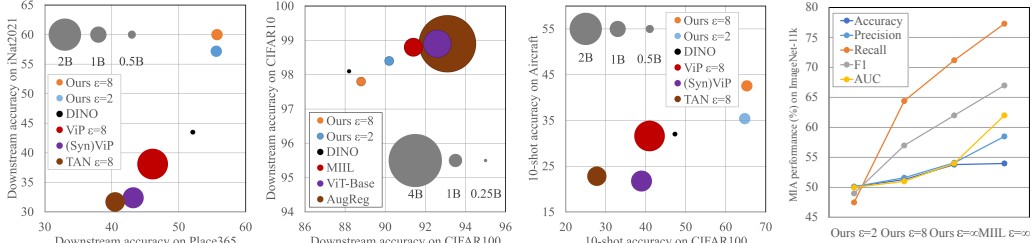

Figure 2: Summary of results in Section 5. First three figures compare the downstream and few-shot performance and the data efficiency (circle's radius proportional to pre-training data size) of the DP pre-trained models; the last figure shows the performance of DP pre-trained models defending against privacy attacks (closer to 0.5 is stronger in defense).

## 1.3 Notions & settings

We denote the mini-batch size as $B$ with index set $\mathcal{B}$, dataset size as $n$, number of training iterations as $T$, iteration index as $t \leq T$, learning rate as $\eta$, model parameters as $\mathbf{w} \in \mathbb{R}^d$, and training samples as $x$. We use $\{k, M, B\}$ for {thousand, million, billion}.

We study DP optimization under a fixed computation budget (with a fixed number of samples), a fixed privacy budget (with a fixed DP guarantee), and limited availability of public data.

## 1.4 Computation budget

We consider a fixed computation budget, so that the training ends when a fixed number of samples $S := BT$ have been processed. As $B$ increases, the number of iterations $T$ decreases linearly, while the per-iteration training time increases almost linearly. This is because foundation models are generally trained with large batch size, which requires the use of distributed learning and gradient accumulation[2]. For example, Vision Transformers (ViT, [27, 74] is trained with $B = 4k$, GPT-3 [6] and LLaMA [77] with $B \approx 2k$, DP-ResNet/ViT with $B = 4k$ in [24] and $B = 1M$ in [58], DP-RoBERTa with $B = 2k$ in [89], and DP-GPT2 with $B = 1k$ in [49].

As a consequence, the batch size $B$ has nearly zero influence on the total training time, leaving its effect only on the convergence speed.

## 1.5 Privacy budget

**Definition 1.1** ([29, 26]). A randomized algorithm $\mathcal{M}$ is $(\varepsilon, \delta)$-DP if, for any two neighboring datasets $\mathcal{S}, \mathcal{S}'$ that differ by one sample and for any event $\mathcal{E}$,

$$\mathbb{P}[\mathcal{M}(\mathcal{S}) \in \mathcal{E}] \leqslant e^{\varepsilon} \mathbb{P}[\mathcal{M}(\mathcal{S}') \in \mathcal{E}] + \delta, \text{ where } \delta < 1/n.$$

We consider a fixed privacy budget $(\epsilon, \delta)$, to be realized through the DP optimizers in (2). Essentially, a noise $\sigma \mathcal{N}(0, \mathbf{I})$ is injected to the gradient at each iteration, whereas the noise magnitude $\sigma(B)$ is determined by privacy accountants such as RDP (default) [59], PRV [34] and GDP [26, 8] (see Figure 3).

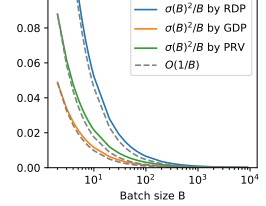

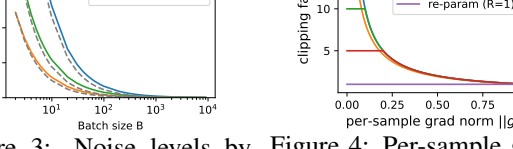

Figure 3: Noise levels by privacy accountants.

Figure 4: Per-sample gradient clipping in (3).

---

[2]The mini-batch of size $B$ (a.k.a. the logical batch size) is divided into $B/b$ micro-batches, where the micro-batch size is $b \ll B$ (a.k.a. the physical or per-GPU batch size, which determines the training speed and memory cost). When $b$ is fixed subject to GPU memory, the per-iteration training time is proportional to the number of micro-batches and thus to $B$.

## 2 Understanding DP training through the lens of Hessian

In this section, we derive the impact of different components in DP training, including the per-sample clipping, the noising, and hyper-parameter choices. To do so, we characterize the per-iteration per-sample loss improvement through the lens of Hessian.

We aim to optimize the generalization loss (equivalently the expected risk) for any differentiable loss function, e.g. the cross entropy, mean squared error, or adversarial loss [33]:

$$\min_{\mathbf{w} \in \mathbb{R}^d} L(\mathbf{w}) = \mathbb{E}_x[L(\mathbf{w}, x)]. \tag{1}$$

We denote the per-sample gradient $\mathbf{g}_i(\mathbf{w}) := \frac{\partial L(\mathbf{w}, x_i)}{\partial \mathbf{w}} \in \mathbb{R}^d$, the oracle gradient $\mathbf{G}(\mathbf{w}) := \frac{\partial \mathbb{E}_x[L(\mathbf{w}, x)]}{\partial \mathbf{w}} \in \mathbb{R}^d$, and the oracle Hessian matrix $\mathbf{H}(\mathbf{w}) := \frac{\partial^2 \mathbb{E}_x[L(\mathbf{w}, x)]}{\partial \mathbf{w}^2} \in \mathbb{R}^{d \times d}$. In Assumption 2.1, samples follow identically and independently (i.i.d.) from some data distribution, with no restriction on the covariance structure $\boldsymbol{\Sigma}(\mathbf{w})$.

**Assumption 2.1.** Per-sample gradients $\mathbf{g}_i(\mathbf{w})$ are i.i.d with

$$\mathbb{E}[\mathbf{g}_i(\mathbf{w})] = \mathbf{G}(\mathbf{w}), \quad \text{Cov}(\mathbf{g}_i(\mathbf{w})) = \boldsymbol{\Sigma}(\mathbf{w}).$$

Consider the general DP optimizers, such as SGD and Adam [46], which update the parameters with the privatized gradient,

$$\mathbf{g} = \frac{\sum_{i \in \mathcal{B}} C_i \mathbf{g}_i + \sigma \mathcal{N}(0, \mathbf{I}_d)}{B} \approx \frac{c \sum_{i \in \mathcal{B}} \mathbf{g}_i + \sigma \mathcal{N}(0, \mathbf{I}_d)}{B}. \tag{2}$$

Here $C_i := C(\|\mathbf{g}_i\|; R)$ is the per-sample clipping factor that restricts the sensitivity of $\sum_i C_i \mathbf{g}_i$ to some constant $R$, i.e. $\|C_i \mathbf{g}_i\| \leq R$. We set $\|C_i \mathbf{g}_i\| \leq 1$ (thus omitting $R$ throughout the paper) following the re-parameterized gradient clipping [24] and the automatic (AUTO) clipping [14], whose $C_i$'s are listed below:

$$C_{i,\text{re-param}} = \min \left\{ \frac{1}{\|\mathbf{g}_i\|}, \frac{1}{R} \right\} \text{ or } C_{i,\text{AUTO}} = \frac{1}{\|\mathbf{g}_i\|}. \tag{3}$$

Figure 4 illustrates the values of $C_i$ under different clipping functions. Note that in (2), we employ a crucial approximation $c \approx \mathbb{E}[C_i]$ so as to unify the formula of DP and non-DP SGD in Remark 2.2. This approximation only holds when the directions of vectors $\sum_i C_i \mathbf{g}_i$ and $\sum_i \mathbf{g}_i$ are very close, i.e., there is little *per-sample clipping bias*. Such approximation is empirically validated in Figure 1, where we observe from the 'SGD' and 'SGD+clipping' curves that the convergence (without noising) is not much influenced by the bias of per-sample gradient clipping.

**Remark 2.2.** Setting $c = 1$ and $\sigma = 0$, the gradient (2) reduces to the standard mini-batch gradient. Hence, the difference between SGD and DP-SGD is characterized by $(\sigma, c)$.

### 2.1 Per-iteration improvement of DP-SGD

Next, we characterize and analyze the per-iteration improvement of DP-SGD through the lens of Hessian, under different choices of hyperparameters and clipping functions: $\mathbf{w}_{t+1} = \mathbf{w}_t - \eta \mathbf{g}_t$. The extension of the analysis to more general optimizers (e.g., DP-Adam) is given in Section 3.4. We are interested in minimizing the second-order Taylor approximation of $L(\mathbf{w} - \eta \mathbf{g})$, which is sufficiently accurate since parameter updates are often quite small [57]. The loss improvement in one iteration can be approximated as:

$$L(\mathbf{w}) - L(\mathbf{w} - \eta \mathbf{g}) \approx \eta \mathbf{G}^\top \mathbf{g} - \frac{\eta^2}{2} \mathbf{g}^\top \mathbf{H} \mathbf{g}.$$

Taking the expectation of the right-hand side, we obtain the expected per-iteration loss improvement (derived in Appendix A.1):

$$\Delta L := \eta \mathbf{G}^\top \mathbb{E}[\mathbf{g}] - \frac{\eta^2}{2} (\text{tr}(\mathbf{H} \text{Cov}(\mathbf{g})) + \mathbb{E}[\mathbf{g}]^\top \mathbf{H} \mathbb{E}[\mathbf{g}]). \tag{4}$$

By applying Assumption 2.1 and (2), we have $\mathbb{E}[\mathbf{g}] = c\mathbf{G}, \quad \text{Cov}(\mathbf{g}) = c^2 \boldsymbol{\Sigma}/B + \sigma^2/B^2$. Substitute to (4), we obtain a quadratic function of $\eta$:

$$\Delta L_{\text{priv}}(\eta, B) := \eta c \mathbf{G}^\top \mathbf{G} - \frac{\eta^2}{2} \left( c^2 \mathbf{G}^\top \mathbf{H} \mathbf{G} + \frac{c^2 \text{tr}(\mathbf{H}\boldsymbol{\Sigma})}{B} + \frac{\sigma^2 \text{tr}(\mathbf{H})}{B^2} \right). \tag{5}$$

We denote the batch size used for SGD and DP-SGD as $B_{\text{non-DP}}$ and $B_{\text{DP}}$, respectively. Then by optimizing the learning rate $\eta$ [3], the per-sample and per-iteration improvement simplifies to

$$\max_{\eta} \Delta L_{\text{priv}}/B_{\text{DP}} = \Delta L_{\text{priv}}^{\star}(B_{\text{DP}}) := \frac{1}{2} \frac{|\mathbf{G}|^4}{B_{\text{DP}} \mathbf{G}^{\top} \mathbf{H} \mathbf{G} + \text{tr}(\mathbf{H}\boldsymbol{\Sigma}) + \sigma^2 \text{tr}(\mathbf{H})/(B_{\text{DP}} \cdot c^2)}. \quad (6)$$

Given that the total number of processed samples is fixed at $S = BT$, (6) can be used as a metric to evaluate the *data efficiency* of DP and non-DP training with different $T, B$ and $(\epsilon, \delta)$.

## 2.2 Per-iteration improvement of vanilla SGD

We can analyze the loss improvement of standard SGD as a sub-case of DP-SGD by substituting $c = 1, \sigma = 0$ into (5), according to Remark 2.2:

$$\Delta L_{\text{pub}} := \eta \mathbf{G}^{\top} \mathbf{G} - \frac{\eta^2}{2} \left( \frac{\text{tr}(\mathbf{H}\boldsymbol{\Sigma})}{B_{\text{non-DP}}} + \mathbf{G}^{\top} \mathbf{H} \mathbf{G} \right), \Delta L_{\text{pub}}^{\star}(B) := \frac{1}{2} \frac{|\mathbf{G}|^4}{B_{\text{non-DP}} \mathbf{G}^{\top} \mathbf{H} \mathbf{G} + \text{tr}(\mathbf{H}\boldsymbol{\Sigma})} \quad (7)$$

We visualize (6), (7), and their individual terms in Figure 5.

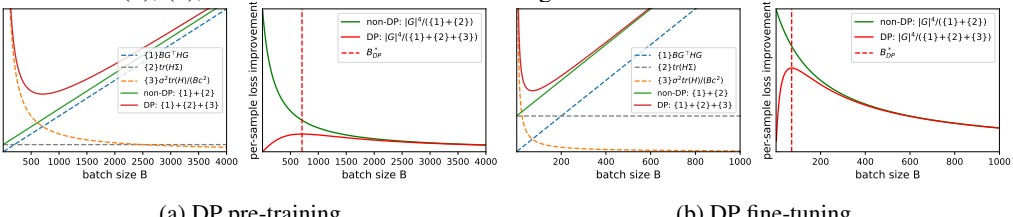

(a) DP pre-training            (b) DP fine-tuning

Figure 5: Illustration of different terms in (6) and (7). Left sub-plots depict the denominators in (6) and (7). Right sub-plots depict the whole terms and optimal batch sizes.

**Implication 2.3** (Better DP mechanism helps). From (6), it is clear that smaller $\sigma$ and larger $c$ (hence larger $C_i$) can help DP training. To reduce $\sigma$, we refer to Section 3.1 for a discussion of methods. For the clipping, under the same sensitivity bound, AUTO clipping [14] gives the largest $C_i$ among all clipping functions (see Figure 4), and therefore is more preferred to use in practice.

**Implication 2.4** (Batch size should be large, but not too large). As visualized by the red solid curves in Figure 5, there exists an optimal batch size (marked in red dashed vertical lines)

$$B_{\text{DP}}^{\star} := \text{argmax}_B \Delta L_{\text{priv}}^{\star}(B) \approx \sqrt{\frac{\sigma^2 \text{tr}(\mathbf{H})}{c^2 \mathbf{G}^{\top} \mathbf{H} \mathbf{G}}}.$$

Compared to previous DP literature, which encourages the batch size to be as a large as possible, our derivation of $B_{\text{DP}}^{\star}$ indicates a sweet pot: while we also support the use of large batch size, we highlight the data inefficiency if $B_{\text{DP}}$ is too large, a case that is often overlooked.

## 3 Impact of per-sample clipping and noising

In this section, we examine the effects of per-sample gradient clipping and noising on the DP training, leveraging the per-iteration per-sample loss improvement derived in Section 2. Specifically, we define and analyze a **"decelerator"** term that characterizes the slowdown by DP optimizers.

Comparing DP-SGD to SGD, we can attribute the slow convergence of DP optimization to the term $\sigma^2 \text{tr}(\mathbf{H})/(Bc^2)$ in (6), which is not present in the standard training (7). We refer to such a term as

$$\textbf{decelerator:} \quad \frac{\sigma^2 \text{tr}(\mathbf{H})}{Bc^2} \approx \frac{\sigma^2 \text{tr}(\mathbf{H})\mathbb{E}|\mathbf{g}_i|^2}{B}, \quad (8)$$

which couples the effects of per-sample gradient clipping and noise through the trace of Hessian. We note that $\text{tr}(\mathbf{H})$ in (8) characterizes the curvature (i.e. sharpness or flatness) of the loss landscape, which strongly correlates with the downstream performance [51, 45, 96, 31].

Next, we discuss how the decelerator impacts the non-DP training, DP pre-training and fine-tuning.

---

[3]The optimal learning rate of DP-SGD cannot be used in practice because (i) the oracle $\mathbf{G}$ and $\mathbf{H}$ are unknown; (ii) it is data-dependent and hence violates the DP guarantee. Such an optimal learning rate is only used to help us understand the best possible per-iteration performance.

## 3.1 No noise, (almost) no deceleration

When $\sigma = 0$ (i.e., no DP noise), the decelerator vanishes and hence (6) reduces to (7), even if the per-sample gradient clipping is used. We empirically verified this in Figure 1 (see blue and black curves), where we see that the difference in convergence with or without clipping is negligible.

Given that DP noise is critical to the convergence speed, we highlight some techniques to reduce $\sigma$ under the same budget of $(\epsilon, \delta)$: the advances in privacy accounting theory can justify smaller noise; algorithms such as LoRA, low-pass and Kalman filters [93, 92] can reduce the effective noise.

## 3.2 DP pre-training can be vulnerable to noise

When $\sigma \neq 0$, the decelerator is non-zero. Therefore, DP training is slowed down by the noise; in Figure 1, SGD with noise (yellow and red curves) has worse performance than SGD without noise (black and blue curves). Furthermore, in pre-training, the decelerator is relatively large in the denominator of (6), i.e., $B\mathbf{G}^\top \mathbf{H}\mathbf{G} + \mathrm{tr}(\mathbf{H}\Sigma) \leq \frac{\sigma^2 \mathrm{tr}(\mathbf{H})}{Bc^2}$ when $B \leq B_{\mathrm{DP}}^\star$ (see left sub-plot of Figure 5(a)), and therefore the slowdown can be significant.

Note that the deceleration issue cannot be resolved by increasing $B$. Although increasing $B$ improves the relative speed of DP convergence in comparison to non-DP, i.e., the decelerator decreases, it hurts the absolute speed since $B\mathbf{G}^\top \mathbf{H}\mathbf{G}$ increases, and thus the loss improvement (6) also worsens (see right sub-plot of Figure 5(a)). Therefore, to design an efficient DP pre-training strategy, we must keep $B$ moderate and reduce the decelerator simultaneously.

## 3.3 DP fine-tuning is robust to noise

Empirical evidence has shown that DP fine-tuning is comparable to (though slightly worse than) the standard non-DP fine-tuning [89, 49, 24, 58, 10, 14, 12], despite that $\sigma \neq 0$. Such a phenomenon implies that comparing public and DP finetuning, we have $\Delta L_{\mathrm{pub}}^\star(B) \approx \Delta L_{\mathrm{priv}}^\star(B)$. That is, the decelerator becomes small after the public pre-training. This is conceptually illustrated in Figure 5(b), where the DP curve is close to the non-DP curve at moderate $B$ during fine-tuning, but not so during pre-training.

To understand the stark contrast between DP fine-tuning and DP pre-training, we plug in the optimal $B_{\mathrm{DP}}^\star = \sqrt{\frac{\sigma^2 \mathrm{tr}(\mathbf{H})}{c^2 \mathbf{G}^\top \mathbf{H}\mathbf{G}}}$ from Implication 2.4 to $\Delta L_{\mathrm{priv}}^\star(B)$. Then, we have the optimal improvement of DP-SGD as

$$\Delta L_{\mathrm{priv}}^\star(B_{\mathrm{DP}}^\star) = \frac{1}{2} \frac{|\mathbf{G}|^4}{2\sqrt{\mathbf{G}^\top \mathbf{H}\mathbf{G} \cdot \sigma^2 \mathrm{tr}(\mathbf{H})/c^2} + \mathrm{tr}(\mathbf{H}\Sigma)}.$$

**Implication 3.1.** Notice that by choosing $B_{\text{non-DP}} = 2B_{\mathrm{DP}}^\star$ in (7), we have that *DP-SGD with the optimal batch size is as fast as the standard SGD with twice the batch size*,

$$\Delta L_{\mathrm{pub}}^\star(2B_{\mathrm{DP}}^\star) = \Delta L_{\mathrm{priv}}^\star(B_{\mathrm{DP}}^\star). \tag{9}$$

Moreover, if $B_{\mathrm{DP}}^\star$ is moderate, then DP-SGD can converge at similar speed to a fast converging SGD that uses a moderate batch size.

**Remark 3.2.** From (7), we observe that non-DP training is data-efficient only if $B_{\text{non-DP}}\mathbf{G}^\top \mathbf{H}\mathbf{G} \ll \mathrm{tr}(\mathbf{H}\Sigma)$. Otherwise, we can decrease $B_{\text{non-DP}}$ to effectively improve $\Delta L_{\mathrm{pub}}^\star$. Therefore, DP training is data-efficient only if $B_{\mathrm{DP}}^\star \approx \frac{1}{2}B_{\text{non-DP}} \ll \frac{\mathrm{tr}(\mathbf{H}\Sigma)}{2\mathbf{G}^\top \mathbf{H}\mathbf{G}}$, which holds in fine-tuning but not in early stage of pre-training[4]. We illustrate the magnitude of the three terms in (6) for pre-training and fine-tuning stages in Figure 6.

In the fine-tuning phase of Figure 6, $\mathrm{tr}(\mathbf{H})$ quickly decreases and so does the decelerator. Hence a moderate $B_{\mathrm{DP}} \approx 100$ can allow fast convergence. However, in the pre-training phase, $\mathrm{tr}(\mathbf{H})$ increases to a large value within 5 epochs and remains for a long time (say epoch 5 to 40) before it decreases again. Consequently, DP convergence is initially fast but only for a short period and overall DP optimization is much slower than non-DP optimization, as shown in Figure 1(a)(c).

---

[4]An intuitive explanation is that $\frac{\mathrm{tr}(\mathbf{H}\Sigma)}{\mathbf{G}^\top \mathbf{H}\mathbf{G}} = \frac{\mathbb{E}[(\mathbf{g}_i - \mathbf{G})^\top \mathbf{H}(\mathbf{g}_i - \mathbf{G})]}{\mathbf{G}^\top \mathbf{H}\mathbf{G}} = \frac{\mathbb{E}(\mathbf{g}_i^\top \mathbf{H}\mathbf{g}_i)}{\mathbb{E}(\mathbf{g}_i)^\top \mathbf{H}\mathbb{E}(\mathbf{g}_i)} - 1$ resembles the variance of per-sample $\mathbf{g}_i$ in the space of $\mathbf{H}$, which decreases as the model learns the common representation.

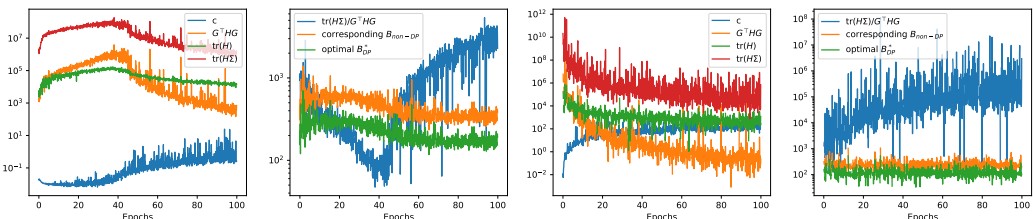

Figure 6: Evolution of terms in (6) and (7) that explains the deceleration of DP optimization, during pre-training (left two) and fine-tuning (right two) ViT-Base on CIFAR100.

## 3.4 Extension to general optimizers

The analysis in the previous two sections can be easily extended to arbitrary optimizers as well as techniques such as weight decay, gradient clipping, and parameter-efficient fine-tuning (PEFT). Let $p$ be an optimizer's post-processor of gradient and consider $\mathbf{w}_{t+1} = \mathbf{w}_t - \eta p(\mathbf{g}_t)$.

For examples, following the notation in Pytorch library [65], we can write Adam and SGD with momentum ($\mu$) and weight decay ($\lambda$),

$$\text{Adam:} \quad p(\mathbf{g}; \mathbf{m}, \mathbf{v}) = \frac{\frac{\beta_1 \mathbf{m} + (1-\beta_1)\mathbf{g}}{1-\beta_1^t}}{\sqrt{\frac{\beta_2 \mathbf{v} + (1-\beta_2)\mathbf{g}^2}{1-\beta_2^t}} + 10^{-8}} \quad \text{SGD}(\mu, \lambda): \quad p(\mathbf{g}; \mathbf{b}, \mathbf{w}) = \mu \mathbf{b} + \mathbf{g} + \lambda \mathbf{w}.$$

Similarly, we can write any PEFT for any optimizer, e.g. PEFT (SGD): $p(\mathbf{g}; \mathbf{M}) = \mathbf{M} \odot \mathbf{g}$ where $\mathbf{M} \in \{0, 1\}$ is an element-wise mask that makes a parameter non-trainable or frozen when $\mathbf{M}_i = 0$.

Specially, we consider optimizers such that $p(\cdot)$ is scale-invariant (i.e. $p(c\mathbf{G}) = p(\mathbf{G})$), such as SignSGD/Adam [4] or normalized SGD/LAMB [61, 55, 87]. Applying Assumption 2.1 and the optimal learning rate, we derive the expected per-sample per-iteration improvement in Implication 3.3, leaving the details Appendix A.5.

**Implication 3.3.** Suppose the post-processor $p(\cdot)$ is scale-invariant. Denote $\mathbf{p} = p(\mathbf{G}), \mathbf{p}' = p'(\mathbf{G})$, the per-sample per-iteration improvement $\max_\eta \Delta L(\eta)/B$ simplifies to

$$\frac{1}{2} \frac{|\mathbf{p}^\top \mathbf{G}|^2}{B\mathbf{p}^\top \mathbf{H}\mathbf{p} + \text{tr}(\mathbf{p}'^\top \mathbf{H}\mathbf{p}'\boldsymbol{\Sigma}) + \sigma^2 \text{tr}(\mathbf{p}'^\top \mathbf{H}\mathbf{p}')/(Bc^2)}$$

Interestingly, similar to the decelerator of DP-SGD (8), the decelerator $\sigma^2 \text{tr}(\mathbf{p}'^\top \mathbf{H}\mathbf{p}')/(Bc^2)$ of these DP optimizers also couples the per-sample gradient clipping, the noise and the Hessian, rendering the theoretical implications from DP-SGD extendable to general DP optimizers.

# 4 Continual pre-training with DP

## 4.1 Necessity of public data in DP pre-training

In this section, we propose the DP continual pre-training strategy and demonstrate that the deceleration by DP can be effectively mitigated by using a certain amount of public data. We consider the mixed data training that uses both public data (with subscript $_0$ for related hyperparameters) and private data (with subscript $_1$). Then SGD becomes

$$\mathbf{g}_{\alpha, t} := \frac{\alpha_t}{B_0} \sum_{j \in \mathcal{B}_0} \mathbf{g}_{j,t} + \frac{(1-\alpha_t)}{B_1} \Big( \sum_{i \in \mathcal{B}_1} C_{i,t} \mathbf{g}_{i,t} + \sigma \mathcal{N}(0, \mathbf{I}_d) \Big).$$

Here $\alpha_t \in [0, 1]$ controls the ratio of privatized and non-privatized gradients, taking different forms by public training (OnlyPublic, $\alpha_t = 1$), private training (OnlyPrivate, $\alpha_t = 0$), DPMD [2], a tunable constant [32, 52], and Sample [30, 44].

Table 2: Summary of $\alpha_t$ by mixed data training methods.

| Ours | DPMD | Sample | OnlyPublic | OnlyPrivate |
|------|------|--------|------------|-------------|
| $\mathbb{I}(t < sT)$ | $1 - \cos \frac{\pi t}{2K}$ | $\frac{n_{\text{pub}}}{n_{\text{pub}} + n_{\text{priv}}}$ | 1 | 0 |

Using the mixed gradient $\mathbf{g}_\alpha$ and following (4), we can show that expected loss improvement is a bivariate quadratic function of $(\eta, \alpha)$ (see Appendix A.3). After minimizing with respect to both variables, we obtain:

$$\alpha^\star = \left( \frac{1}{c} \frac{\text{tr}(\mathbf{H}\boldsymbol{\Sigma}) \cdot B_1/B_0}{\text{tr}(\mathbf{H}\boldsymbol{\Sigma}) + \sigma^2 \text{tr}(\mathbf{H})/(B_1 c^2)} + 1 \right)^{-1} \tag{10}$$

**Remark 4.1.** We see that the optimal choice in (10) gives $\alpha^\star \neq 0$, which indicates that $\Delta L^\star_{\text{OnlyPrivate}} < \Delta L^\star_{\text{mixed}}$, i.e. the public data helps. On the other hand, the fact that optimal $\alpha^\star \neq 1$ also indicates that $\Delta L^\star_{\text{OnlyPublic}} < \Delta L^\star_{\text{mixed}}$, i.e. the private data helps.

## 4.2 DP continual pre-training strategy overview

Motivated by Section 3 and Remark 4.1, we propose a two-phase DP continual pre-training strategy in Appendix D: public pre-training for $sT$ steps followed by private continual pre-training for $(1 - s)T$ steps, which is equivalent to setting $\alpha_t = \mathbb{I}(t < sT)$ and the constant $0 < s < 1$ controls the portion of steps of public training.

**Remark 4.2.** Although (10) suggests that the optimal $\alpha_t \in (0, 1)$, we only set binary $\alpha_t \in \{0, 1\}$ so as to not implement two data loaders and two back-propagation mechanisms simultaneously (one for DP and one for standard). Note the methods in Table 2 are difficult to scale and not yet openly implemented on distributed systems due to memory and synchronization issues.

We highlight that DP continual pre-training is as distinct from DP fine-tuning as their non-DP counter-parts, despite both methods extending the learning from a pre-trained phase: the continual pre-training learns common knowledge without adapting to a specific task and serves a richer foundation for many downstream tasks (see more discussion in Appendix C). In fact, we show in Appendix A.7 that the loss improvement of DP continual pre-training can be almost as fast as non-DP pre-training on the full data (FullyPublic), thus closing the utility gap.

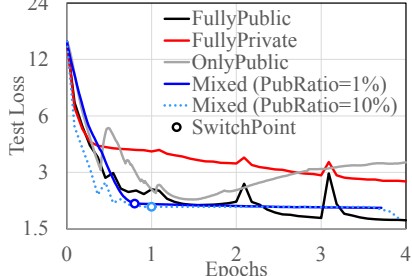

Table 3: Optimization by different training strategies.

|  | public data | private data | privacy |
|---|---|---|---|
| OnlyPublic | SGD | not used | Yes |
| OnlyPrivate | not used | DP-SGD | Yes |
| FullyPublic | SGD | SGD | No |
| FullyPrivate | DP-SGD | DP-SGD | Yes |
| Mixed (our $\mathbf{g}_\alpha$) | SGD | DP-SGD | Yes |

Figure 7: Pre-training GPT2-small on CodeParrot with different pre-training strategies ($\epsilon = 8$ if Mixed or FullyPrivate).

The switching (i.e. selecting $s$) can be automatically determined by public statistics without using any DP budget. For example, we can use early stopping based on the loss or accuracy, switching after the metrics stop improving (see Figure 7). Alternatively, we can monitor $B^\star_{\text{DP}}$ and switch when it drops to a moderate value so that the DP training converges fast by (9).

# 5  DP vision foundation models on ImageNet

We leverage the DP continual pre-training strategy discussed in Section 4 to train vision transformers [27] on ImageNet datasets. We use ImageNet-1k (1.3M images, 1k classes; [25]) for public pre-training, then ImageNet-11k (formally known as ImageNet-21k-P[5], 11M images, 11k classes; [70]) for private pre-training. Notice that ImageNet-11k/21k is significantly harder to learn, with SOTA accuracy $\approx 47\%$ [70, 74] as compared to $> 85\%$ for ImageNet-1k. We apply data augmentation including random flipping, contract shift, rotation, and resizing to $224 \times 224$.

We evaluate our DP ViT on upstream and downstream tasks (including few-shot), achieving high accuracy under low computation budget (compared to existing DP pre-trained models in Table 4). In short, we show that unlocking more data (90% of full ImageNet), which cannot be used by DINO due to privacy concern, is significantly beneficial.

---

[5]ImageNet-11k improves the dataset quality and thus the resulting models (see Table 3 in [70]), by removing the infrequent classes from the full ImageNet-21k, thus remaining 78% of its 14M images and 48% of 21.8k classes.

Table 4: Pre-training strategies of models. Standard non-DP training is marked in black; DP training is in green. † indicates self-supervised without using the labels. "Images ×" is the total number of images used (dataset size×epochs). "Non-privacy" means no DP guarantee on a subset of training data due to the non-DP pre-training phase.

| | reference | model | pre-training | continual training | non-privacy | images × |
|---|---|---|---|---|---|---|
| TAN | [71] | | ImageNet-1k | — | — | 1.2B |
| (Syn)ViP | [90] | ViT-Base | Shaders21k† | | Shaders21k | 1.3B |
| ViP | [90] | ViT-Base | Shaders21k† | LAION400M† | Shaders21k | 1.9B |
| DINO | [19] | ViT-Base | ImageNet-1k† | — | ImageNet-1k | 0.3B |
| Ours | This work | ViT-Base | ImageNet-1k† | ImageNet-11k | ImageNet-1k | 0.7B |
| MIIL | [70] | ViT-Base | ImageNet-1k | ImageNet-11k | ImageNet-11k | 1.4B |
| Original | [27] | ViT-Base | ImageNet-21k | ImageNet-1k | ImageNet-21k | 1.2B |
| AugReg | [74] | ViT-Base | ImageNet-1k | ImageNet-21k | ImageNet-21k | 4.3B |
| NFnet-JFT | [24] | ResNet-50 | JFT300M | — | JFT300M | 4.0B |

## 5.1 Training strategy

For the public pre-training, we follow the self-supervised learning by [19] (self-distillation with no labels, or DINO), whereas the private continual pre-training is a supervised learning following [70][6]. We employ AdamW optimizer with batch size $B = 4096$ and learning rate $\eta = 0.0002$ set by the line search. Our training strategy is similar to a concurrent work [90], with critical differences highlighted in Appendix C.

When automatically switching from public to private pre-training, the classification head (the last layer) is re-initialized because the number of classes is different in the two pre-training phases: we switch from 1k classes to 11k classes. This switching is triggered by early stopping. In the continual pre-training phase, we train with DP linear probing for 10 epochs then with DP full-parameter training for 20 epochs, with a total $\approx$100k training steps. This strategy achieves an upstream accuracy 41.5% on ImageNet-11k by our ViT-Base under $\epsilon = 8$, and 39.8% under $\epsilon = 2$.

## 5.2 Algorithm implementation

We employ `fastDP` library to apply the DP-AdamW with automatic per-sample clipping function [14] and layer-wise clipping style [9]. Specifically, the DP optimization is under the multi-GPU distributed system, using DP-ZeRO [7] and mixed-precision training, so as to enjoy the same training speed and memory efficiency as the non-DP training. We calibrate the DP noise using the improved Renyi accountant.

## 5.3 Downstream performance

Our DP pre-training learns highly transferable representations, demonstrated through the strong performance on a list of downstream datasets. We summarize in Table 4 a number of models from previous literature for comparison. We mark **DP pre-trained models** in green and leave **non-DP models** in black. The most informative baselines are DINO and MIIL, since our DP models continue the pre-training from DINO, following a strategy similar to MIIL.

In Table 5, we compare different pre-training strategies (all non-DP except ours) leveraging the same dataset and same model architecture – ViT-Base (86M param). Our evaluation shows that DP pre-trained models achieve high downstream accuracy under standard and non-DP fine-tuning: 98.4% on CIFAR10, 90.2% on CIFAR100, 86.5% on Food101 and 96.8% on SVHN. Our DP continual pre-training clearly improves upon DINO, with $+0.3 \sim 2.0\%$ on accuracy, and is comparable to the non-DP pre-trained MIIL that uses twice the data size (1.4B v.s. our 0.7B).

In Table 6, when the downstream tasks (non-DP) are more challenging with only a few data samples to learn from, our DP model substantially outperforms previous DP pre-trained models across all settings, for example, by $+19 \sim 38\%$ on CIFAR100 when compared to ViP and TAN. We attribute the success to the high quality of pre-training data, i.e. ImageNet-1k/11k, in contrast to Shaders (by comparing DINO to (Syn)ViP) and LAION (by comparing the improvement from DINO to ours and from (Syn)ViP to ViP).

---

[6]We observe that, using AugReg (a supervised pre-training strategy) for public pre-training on ImageNet-1k and/or for private pre-training on ImageNet-21k gives very similar result.

Table 5: Standard/DP fine-tuning accuracy with the same architecture (ViT-Base) and pre-training dataset (ImageNet-21k) up to subsampling and preprocessing. Number of processed images by each model is indicated in the parenthesis.

| Fine-tuning | CIFAR10 | | | CIFAR100 | | | Food101 | | | SVHN | | |
|---|---|---|---|---|---|---|---|---|---|---|---|---|
| Pre-training | non-DP | $\epsilon=8$ | $\epsilon=2$ | non-DP | $\epsilon=8$ | $\epsilon=2$ | non-DP | $\epsilon=8$ | $\epsilon=2$ | non-DP | $\epsilon=8$ | $\epsilon=2$ |
| DINO (0.3B) | 98.1 | 97.2 | 97.0 | 88.2 | 84.7 | 82.7 | 85.2 | 77.2 | 73.5 | 96.2 | 91.7 | 90.3 |
| Ours$_{\epsilon=2}$ (0.7B) | 97.8 | 96.6 | 96.1 | 88.8 | 83.1 | 81.1 | 84.8 | 75.5 | 72.5 | 96.3 | 91.3 | 90.1 |
| Ours$_{\epsilon=8}$ (0.7B) | 98.4 | 97.2 | 96.9 | 90.2 | 85.0 | 82.8 | 86.5 | 78.4 | 75.3 | 96.8 | 92.5 | 91.3 |
| MIIL (1.4B) | 98.8 | 98.5 | 98.2 | 91.4 | 90.9 | 89.2 | 87.2 | 84.5 | 83.0 | 96.8 | 93.3 | 92.0 |
| ViT_base (1.2B) | 98.9 | 98.3 | 98.1 | 92.6 | 89.9 | 88.2 | 89.4 | 85.5 | 83.1 | 96.9 | 93.5 | 92.5 |
| AugReg (4.3B) | 98.9 | 98.8 | 98.5 | 93.1 | 91.2 | 90.4 | 90.2 | 87.6 | 85.7 | 96.9 | 93.8 | 92.5 |

Table 6: Few-shot accuracy of DP pre-trained models (TAN, ViP and ours) and their non-DP initialization.

| | Aircraft (10-shot) | Aircraft (20-shot) | CIFAR100 (10-shot) | CIFAR100 (30-shot) | fine-tune epochs |
|---|---|---|---|---|---|
| TAN$_{\epsilon=8}$ | 22.84 | 37.93 | 27.78 | 42.35 | 200 |
| (Syn)ViP | 21.79 | 46.85 | 38.96 | 55.84 | 200 |
| ViP$_{\epsilon=8}$ | 31.62 | 53.05 | 40.95 | 57.52 | 200 |
| DINO | 32.04 | 45.61 | 47.31 | 66.92 | 100 |
| Ours$_{\epsilon=2}$ | 36.42 | 48.27 | 64.74 | 74.62 | 100 |
| Ours$_{\epsilon=8}$ | **42.57** | **57.15** | **65.26** | **76.38** | 100 |

Table 7: Linear-probing accuracy (non-DP) of pre-trained models, except "full" indicating full-parameter.

| | ImageNet-1k | Places365 | iNat2021 | fine-tune |
|---|---|---|---|---|
| # images | 1M | 1.8M | 2.7M | epochs |
| TAN$_{\epsilon=8}$ | 49.0 | 40.5 | 31.7 | 90 / 90 / 90 |
| (Syn)ViP | 49.8 | 43.2 | 32.4 | 90 / 90 / 90 |
| ViP$_{\epsilon=8}$ | 55.7 | 46.1 | 38.1 | 90 / 90 / 90 |
| DINO | ~~76.1~~ | 52.1 | 43.5 | 8 / 5 / 10 |
| Ours$_{\epsilon=2}$ | ~~76.2~~ | 52.5 | 46.5 | 8 / 5 / 10 |
| Ours$_{\epsilon=8}$ | ~~77.9~~ | 53.0 | 49.1 | 8 / 5 / 10 |
| Ours$_{\epsilon=2}$(full) | ~~78.0~~ | 55.6 | 57.2 | 8 / 5 / 10 |
| Ours$_{\epsilon=8}$(full) | ~~78.5~~ | 55.7 | 60.0 | 8 / 5 / 10 |
| NFnet-JFT | 74.1 | 54.5 | — | 10 / 26 / — |

In Table 7[7], we extend the evaluation of full fine-tuning and linear probing to SOTA non-DP baselines and to million-image scale Our DP model achieves 55.7% (+9.6% over ViP) on Places365 with 1.8M images and 60.0% (+21.9% over ViP) on iNat2021 with 2.7M images. The current non-DP SOTA is 57-60% on Places365 [24, 27] and about 64% on iNat2021 [80, 60] after pre-training on $2.7 \sim 4$B images. This showcases the effectiveness of DP pre-training as our models only leverage 0.7B images.

## 5.4 Privacy protection

We employ a white-box membership inference attack (MIA) to evaluate the data protection by our DP pre-training: 1) for each image in ImageNet-11k, we compute its output logits and loss, which serves as the feature of the MIA dataset; 2) we randomly select $50\%$ of the testing images and the same number of training images ($522,496$ samples) as the MIA test set, and the rest data as MIA train set; 3) we label the training images as class "1" and testing images as class "0". This creates the MIA dataset with 11k features and binary labels.

We fit a logistic regression with the MIA training set to classify whether an image belongs to the training set of ImageNet-11k (class "1") or not. We report the results on the MIA testing set in Table 8, showing the effectiveness of DP protection when $\epsilon \leq 8$.

Table 8: Membership inference attack results. Values closer to $0.5$ indicate better privacy protection.

| | Accuracy | Precision | Recall | F1 | AUC |
|---|---|---|---|---|---|
| Ours$_{\epsilon=2}$ | **50.1%** | **50.1%** | **47.5%** | **0.49** | **0.50** |
| Ours$_{\epsilon=8}$ | 51.3% | 51.6% | 64.4% | 0.57 | 0.51 |
| Ours$_{\epsilon=\infty}$ | 53.8% | 54.1% | 71.2% | 0.62 | 0.54 |
| MIIL | 54.0% | 58.5% | 77.3% | 0.67 | 0.62 |

## 6 Discussion

In this paper, we conduct an insightful and unified convergence analysis on DP optimization. Specifically, we identify the decelerator (8) of DP training as a result of the per-sample gradient clipping, the noise and the Hessian, which can be significantly mitigated by a small amount ($< 10\%$) of public training. Consequently, we propose DP continual pre-training that is almost as accurate and implementable as the fully public pre-training.

---

[7]For ImageNet-1k, we strike through some results because this dataset was used in the pre-training. It is only meaningful to compare our models to DINO and observe the benefit of continual pre-training, but not to others.

## Acknowledgement

The work of Xinwei Zhang was partially done while interning at Amazon. Mingyi Hong holds concurrent appointments as an Amazon Scholar and as a faculty at the University of Minnesota. This paper describes their work performed at Amazon.

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

# A Derivation and proofs

## A.1 Derivation of Equation (4)

$$\Delta L := \eta \mathbf{G}^\top \mathbb{E}[\mathbf{g}] - \frac{\eta^2}{2}\mathbb{E}[\mathbf{g}^\top \mathbf{H}\mathbf{g}]$$

$$\stackrel{(a)}{=} \eta \mathbf{G}^\top \mathbb{E}[\mathbf{g}] - \frac{\eta^2}{2}\text{tr}\left(\mathbf{H}\mathbb{E}[\mathbf{g}\mathbf{g}^\top]\right)$$

$$\stackrel{(b)}{=} \eta \mathbf{G}^\top \mathbb{E}[\mathbf{g}] - \frac{\eta^2}{2}\text{tr}\left(\mathbf{H}\text{Cov}(\mathbf{g})\right) + \text{tr}\left(\mathbf{H}\mathbb{E}[\mathbf{g}]\mathbb{E}[\mathbf{g}]^\top\right)$$

$$\stackrel{(c)}{=} \eta \mathbf{G}^\top \mathbb{E}[\mathbf{g}] - \frac{\eta^2}{2}\left(\text{tr}\left(\mathbf{H}\text{Cov}(\mathbf{g})\right) + \mathbb{E}[\mathbf{g}]^\top \mathbf{H}\mathbb{E}[\mathbf{g}]\right)$$

where $(a)$ uses equations (13) and then (12) to the second term; $(b)$ separates the expectation of $\mathbf{g}\mathbf{g}^\top$ into its mean squared and covariance and uses (11); $(c)$ uses equations (12) and then (13) to the last term.

**Property of the trace:**

$$\text{tr}(\mathbf{A} + \mathbf{B}) = \text{tr}(\mathbf{A}) + \text{tr}(\mathbf{B}), \quad \text{tr}(c\mathbf{A}) = c\text{tr}(\mathbf{A}), \qquad \text{Linearity}, \qquad (11)$$

$$\text{tr}(\mathbf{A}\mathbf{B}) = \text{tr}(\mathbf{B}\mathbf{A}), \qquad\qquad \text{Trace of product}, \qquad (12)$$

$$\text{tr}(a) = a, \qquad\qquad \text{Trace of a number}. \qquad (13)$$

## A.2 Explanation for Implication 2.4

We will leverage GDP to validate that $\sigma(B)^2/B \approx O(1/B)$, given that most of other privacy accountants are numerical and hard to interpret. We note that $\mu$-GDP has an one-to-one mapping with the $(\varepsilon, \delta)$-DP:

$$\delta = \Phi(-\frac{\epsilon}{\mu} + \frac{\mu}{2}) + \Phi(-\frac{\epsilon}{\mu} - e^\epsilon \frac{\mu}{2})$$

where $\Phi$ is the normal cumulative distribution function.

**Lemma A.1.** *Given an iterative algorithm with $\ell_2$ sensitivity $1$ at each iteration, which uniformly samples the data in dataset of size $n$ with ratio $\frac{B}{n}$, by injecting Gaussian noise $\mathcal{N}(0, \sigma^2\mathbf{I})$ to the output of the algorithm at each iteration, it satisfies $\mu$-GDP with*

$$\mu = \frac{B}{n}\sqrt{T(e^{1/\sigma(B)^2} - 1)} = \sqrt{BS(e^{1/\sigma(B)^2} - 1)}/n,$$

*where $S$ denotes the fixed computation budget.*

The proof is equivalent to that in Section 2.4 in [26].

By Taylor expansion over $B$,

$$\sigma(B)^2 = \frac{1}{\log(\frac{\mu^2 n^2}{BS} + 1)} = \frac{BS}{\mu^2 n^2} + \frac{1}{2} + O(\frac{1}{B}) \longrightarrow \frac{\sigma(B)^2}{B} = \frac{S}{\mu^2 n^2} + \frac{1}{2B} + o(\frac{1}{B})$$

In the pre-training regime, $S/n$ is the number of epochs (usually between 1 and 300) and the sample size $n$ is huge, i.e. $n > 10^7$ for ImageNet and $n > 10^{12}$ for large language model training. Hence the first term is negligible and $\sigma(B)^2/B \approx 0.5/B$.

## A.3 Proof of results in Section 3.3

*Proof.* Leveraging (2), we have

$$\mathbb{E}(\mathbf{g}) = \alpha \mathbf{G} + (1 - \alpha)c\mathbf{G}, \ \text{Cov}(\mathbf{g}) = \alpha^2 \frac{\mathbf{\Sigma}}{B_0} + (1 - \alpha)^2(\frac{c^2\mathbf{\Sigma}}{B_1} + \frac{\sigma^2\mathbf{I}}{B_1^2})$$

Substituting in the mixed gradient $\mathbf{g}_\alpha$ and following (4), we can write down the expected improvement of mixed data training:

$$\Delta L_{\text{mixed}} = \eta(\alpha + (1-\alpha)c)\mathbf{G}^\top\mathbf{G} - \frac{\eta^2(1-\alpha)^2\sigma^2}{2B_1^2}\text{tr}(\mathbf{H})$$
$$- \frac{\eta^2}{2}(\frac{\alpha^2}{B_0} + \frac{(1-\alpha)^2}{B_1}c^2)\text{tr}(\mathbf{H\Sigma})$$
$$- \frac{\eta^2}{2}(\alpha + (1-\alpha)c)^2\mathbf{G}^\top\mathbf{H}\mathbf{G},$$

which is a bivariate quadratic function $\Delta L_{\text{mixed}}$ to be minimized over $(\eta, \alpha)$. To make the presentation easier, we denote $\eta_0 = \eta\alpha, \eta_1 = \eta(1-\alpha)$, and the above equation can be rewritten as:

$$\Delta L_{\text{mixed}} = (\eta_1 c + \eta_0)\mathbf{G}^\top\mathbf{G} - \frac{1}{2}\frac{\eta_1^2\sigma^2}{B_1^2}\text{tr}(\mathbf{H}) - \frac{1}{2}(\frac{\eta_1^2}{B_1}c^2 + \frac{\eta_0^2}{B_0})\text{tr}(\mathbf{H\Sigma}) - \frac{1}{2}(\eta_1 c + \eta_0)^2\mathbf{G}^\top\mathbf{H}\mathbf{G}.$$

Note that if we set $\eta_1 = 0$ or $\eta_0 = 0$, we essentially train the model with only public data or private data, respectively.

can be written as a bivariate quadratic function over $(\eta_0, \eta_1)$:

$$\Delta L_{\text{mixed}} = -(A\eta_0^2 + B\eta_1^2 + C\eta_0 + D\eta_1 + E\eta_0\eta_1)$$

in which

$$A = \frac{1}{2}\mathbf{G}^\top\mathbf{H}\mathbf{G} + \frac{1}{2}\frac{\text{tr}(\mathbf{H\Sigma})}{B_0},$$
$$B = \frac{1}{2}c^2\mathbf{G}^\top\mathbf{H}\mathbf{G} + \frac{1}{2}c^2\frac{\text{tr}(\mathbf{H\Sigma})}{B_1} + \frac{1}{2}\frac{\sigma^2\text{tr}(\mathbf{H})}{B_1^2},$$
$$C = -\mathbf{G}^\top\mathbf{G},$$
$$D = -c\mathbf{G}^\top\mathbf{G},$$
$$E = c\mathbf{G}^\top\mathbf{H}\mathbf{G}.$$

The maximizer is

$$\eta_0 = -\frac{2BC - DE}{4AB - E^2}, \eta_1 = -\frac{2AD - CE}{4AB - E^2}.$$

Hence

$$\frac{1-\alpha^*}{\alpha^*} = \frac{2AD - CE}{2BC - DE} \implies \alpha^* = 1/\left(\frac{2AD - CE}{2BC - DE} + 1\right)$$

Finally

$$\alpha^* = \left(\frac{c\,\text{tr}(\mathbf{H\Sigma})\mathbf{G}^\top\mathbf{G}/B_0}{c^2\text{tr}(\mathbf{H\Sigma})\mathbf{G}^\top\mathbf{G}/B_1 + \sigma^2\text{tr}(\mathbf{H})\mathbf{G}^\top\mathbf{G}/B_1^2} + 1\right)^{-1} = \left(\frac{1}{c}\frac{\text{tr}(\mathbf{H\Sigma})\cdot B_1/B_0}{\text{tr}(\mathbf{H\Sigma}) + \sigma^2\text{tr}(\mathbf{H})/(B_1 c^2)} + 1\right)^{-1}$$

$\square$

### A.4 Explanation of Remark 3.2

From (6), the optimal batch size for DP-SGD is

$$\arg\max_B \frac{1}{2}\frac{|\mathbf{G}|^4}{B\mathbf{G}^\top\mathbf{H}\mathbf{G} + \text{tr}(\mathbf{H\Sigma}) + \sigma^2\text{tr}(\mathbf{H})/(Bc^2)}$$
$$= \arg\min_B B\mathbf{G}^\top\mathbf{H}\mathbf{G} + \text{tr}(\mathbf{H\Sigma}) + \sigma^2\text{tr}(\mathbf{H})/(Bc^2) \approx \sqrt{\frac{\sigma^2\text{tr}(\mathbf{H})}{c^2\mathbf{G}^\top\mathbf{H}\mathbf{G}}}$$

which minimizes (6) to

$$\text{DP-SGD}(B_{\text{DP}}^*) \longrightarrow \frac{1}{2}\frac{|\mathbf{G}|^4}{2\sqrt{\mathbf{G}^\top\mathbf{H}\mathbf{G}\cdot\sigma^2\text{tr}(\mathbf{H})/c^2} + \text{tr}(\mathbf{H\Sigma})}$$

where $B_{\text{DP}}^* := \sqrt{\frac{\sigma^2 \text{tr}(\mathbf{H})}{c^2 \mathbf{G}^\top \mathbf{H} \mathbf{G}}}$.

Note this is equivalent to applying $B_{\text{non-DP}} := 2\sqrt{\frac{\sigma^2 \text{tr}(\mathbf{H})}{c^2 \mathbf{G}^\top \mathbf{H} \mathbf{G}}} = 2B_{\text{DP}}^*$ on (7),

$$\text{SGD}(B_{\text{non-DP}}) \equiv \text{SGD}(2B_{\text{DP}}^*) \equiv \text{DP-SGD}(B_{\text{DP}}^*).$$

We emphasize that generally

$$\text{SGD}(2B_{\text{DP}}) \not\equiv \text{DP-SGD}(B_{\text{DP}}).$$

## A.5   Loss improvement of general DP optimizers – Implication 3.3

From $\mathbf{w}_{t+1} = \mathbf{w}_t - \eta p(\mathbf{g}_t)$, the expected per-iteration loss improvement becomes

$$\Delta L = \eta \mathbf{G}^\top \mathbb{E}[p(\mathbf{g})] - \frac{\eta^2}{2}\left(\text{tr}\left(\mathbf{H}\text{Cov}(p(\mathbf{g}))\right) + \mathbb{E}[p(\mathbf{g})]^\top \mathbf{H} \mathbb{E}[p(\mathbf{g})]\right). \qquad (14)$$

Applying Assumption 2.1 and delta method, with $\mathbf{G} = \frac{\partial L}{\partial \mathbf{w}}$, we have

$$\sqrt{B}(p(\mathbf{g}) - p(c\mathbf{G})) = \sqrt{B}p'(c\mathbf{G}) \cdot (\mathbf{g} - c\mathbf{G}) + o_p(1) \to \mathcal{N}\left(0, p'(c\mathbf{G})[c^2\mathbf{\Sigma} + \sigma^2/B]p'(c\mathbf{G})^\top\right)$$

where $p'(c\mathbf{G}) \equiv \frac{\partial p(c\mathbf{G})}{\partial c\mathbf{G}} \in \mathbb{R}^{d \times d}$. Hence

$$\mathbb{E}(p(\mathbf{g})) \approx p(c\mathbf{G}), \text{Cov}(p(\mathbf{g})) \approx p'(c\mathbf{G}) \cdot \left(c^2\mathbf{\Sigma}/B + \sigma^2/B^2\right) \cdot p'(c\mathbf{G})^\top$$

Next, the expected improvement contributed by one sample is a quadratic function of $\eta$:

$$\Delta L/B \approx \left(\eta p(c\mathbf{G})^\top \mathbf{G} - \frac{\eta^2}{2}\frac{\sigma^2 \text{tr}[p'(c\mathbf{G})^\top \mathbf{H} p'(c\mathbf{G})]}{B^2}\right.$$
$$\left. - \frac{\eta^2}{2}\frac{c^2 \text{tr}[p'(c\mathbf{G})^\top \mathbf{H} p'(c\mathbf{G})\mathbf{\Sigma}]}{B} - \frac{\eta^2}{2}p(c\mathbf{G})^\top \mathbf{H} p(c\mathbf{G})\right)/B$$

Applying the optimal learning rate, the optimal per-sample loss improvement $\Delta L/B$ at each iteration simplifies to

$$\frac{1}{2}\frac{|p(c\mathbf{G})^\top \mathbf{G}|^2}{Bp(c\mathbf{G})^\top \mathbf{H} p(c\mathbf{G}) + c^2 \text{tr}(p'(c\mathbf{G})^\top \mathbf{H} p'(c\mathbf{G})\mathbf{\Sigma}) + \sigma^2 \text{tr}(p'(c\mathbf{G})^\top \mathbf{H} p'(c\mathbf{G}))/B}$$

In the special case that $p$ is scale-invariant, e.g. in adaptive optimizers like Adam or in SignSGD, we get $p'(c\mathbf{G}) = \frac{\partial p(c\mathbf{G})}{\partial c\mathbf{G}} = \frac{\partial p(\mathbf{G})}{\partial \mathbf{G}}\frac{\partial \mathbf{G}}{\partial c\mathbf{G}} = p'(\mathbf{G})/c$ and thus

$$\frac{1}{2}\frac{|p(\mathbf{G})^\top \mathbf{G}|^2}{Bp(\mathbf{G})^\top \mathbf{H} p(\mathbf{G}) + \text{tr}(p'(\mathbf{G})^\top \mathbf{H} p'(\mathbf{G})\mathbf{\Sigma}) + \sigma^2 \text{tr}(p'(\mathbf{G})^\top \mathbf{H} p'(\mathbf{G}))/(Bc^2)}$$

## A.6   Improvement of performance measures other than the optimization loss

We now consider two extended cases when the optimization loss is different to the performance measures. For example, the model may be trained via the cross-entropy loss but measured on 0-1 accuracy.

| optimization | performance | | example |
|:---:|:---:|:---:|:---:|
| $L$ | $L$ | vanilla | $L$ is cross-entropy |
| $L$ | $L_{\text{other}}$ | vanilla | $L$ is cross-entropy; $L_{\text{other}}$ is BLEU or accuracy |
| $L_{\text{mod}}$ | $L$ | adversarial training | $L_{\text{mod}}$ is adversarial loss; $L$ is cross-entropy |

Table 9: DP/non-DP optimization when the optimization loss is different to the performance measures.

For the first case, we analyze many performance measures of DP models (denoted as $L_{\text{other}}$) beyond the optimization loss $L$. For instance, foundation models trained on cross-entropy loss can be evaluated on classification accuracy, F1 score, BLEU [64], ROGUE [50], fairness [37], calibration [35], adversarial robustness, etc.

We demonstrate that our analysis in previous sections indeed generalizes: equivalent to (5), we have

$$\Delta L_{\text{other}}(\eta) = \eta \mathbf{G}_{\text{other}}^\top \mathbb{E}[\mathbf{g}] - \frac{\eta^2}{2} \mathbb{E}[\mathbf{g}^\top \mathbf{H}_{\text{other}} \mathbf{g}]$$

where $\mathbf{G}_{\text{other}}, \mathbf{H}_{\text{other}}$ are the oracle gradient and Hessian of $L_{\text{other}}$. The per-sample per-iteration improvement, $\max_\eta \Delta L_{\text{other}}/B$, simplifies to

$$\frac{1}{2} \frac{|\mathbf{G}_{\text{other}}^\top \mathbf{G}|^2}{B \mathbf{G}^\top \mathbf{H}_{\text{other}} \mathbf{G} + \text{tr}(\mathbf{H}_{\text{other}} \mathbf{\Sigma}) + \sigma^2 \text{tr}(\mathbf{H}_{\text{other}})/(Bc^2)},$$

with the decelerator being $\sigma^2 \text{tr}(\mathbf{H}_{\text{other}})/(Bc^2)$ in place of (8). As implied by Section 3.3, we expect the public pre-training also mitigates this decelerator, so that well-trained DP models and non-DP models should be equally performant. Empirically speaking, DP fine-tuning has shown to be as accurate [49, 24], adversarially robust [15], calibrated [11], and fair [5] as the standard fine-tuning.

For the second case, e.g. adversarial, sharpness-aware, fairness-aware or calibration-aware training, the optimization is on the modified loss $\frac{\partial L_{\text{mod}}}{\partial \mathbf{w}}$ but the performance is measured on the vanilla loss $\frac{\partial L}{\partial \mathbf{w}}$. For instance, we want an adversarially trained model to be sufficiently accurate. To be specific, FGSM and PGD ($L_2/L_\infty$ perturbation with norm $\rho$) lead to the modified loss

$$L_{\text{mod}} := \max_{||\xi|| \leq \rho} \mathbb{E}_x[L(\mathbf{w}, x + \xi)], \mathbf{G}_{\text{mod}} = \frac{\partial L_{\text{mod}}}{\partial \mathbf{w}}.$$

DP adversarial training applies on

$$\mathbf{p} = \frac{1}{B}\left(\sum_i C_i \mathbf{g}_{\text{mod},i} + \sigma \mathcal{N}(0, \mathbf{I}_d)\right) \approx \frac{1}{B}\left(c \sum_i \mathbf{g}_{\text{mod},i} + \sigma \mathcal{N}(0, \mathbf{I}_d)\right)$$

where per-sample gradient is $\mathbf{g}_{\text{mod},i} := \frac{\partial \max_{||\xi|| \leq \rho} L(\mathbf{w}, x_i + \xi)}{\partial \mathbf{w}}$. Note that the modified gradient may be post-processed by optimizers like AdamW. To be clear, only in this section, we write $\mathbf{p} = \frac{\partial L_{\text{mod}}}{\partial \mathbf{w}}$ and omit the post-processing of the optimizer (i.e. we only show for SGD).

Next, the expected per-iteration loss improvement becomes

$$\Delta L = \eta \mathbf{G}^\top \mathbb{E}[\mathbf{p}] - \frac{\eta^2}{2}\left(\text{tr}\left(\mathbf{H}\text{Cov}(\mathbf{p})\right) + \mathbb{E}[\mathbf{p}]^\top \mathbf{H}\mathbb{E}[\mathbf{p}]\right). \tag{15}$$

Applying Assumption 2.1, we have

$$\mathbb{E}[\mathbf{p}] = c\mathbf{G}_{\text{mod}}, \text{Cov}(\mathbf{p}) = c^2 \mathbf{\Sigma}_{\text{mod}}/B + \sigma^2/B^2$$

Hence, the expected per-iteration improvement contributed by one sample is a quadratic function of $\eta$:

$$\Delta L/B := (\eta c \mathbf{G}^\top \mathbf{G}_{\text{mod}} - \frac{\eta^2}{2}\frac{\sigma^2 \text{tr}(\mathbf{H})}{B^2} - \frac{\eta^2}{2}\frac{c^2 \text{tr}(\mathbf{H}\mathbf{\Sigma}_{\text{mod}})}{B} - \frac{\eta^2}{2}c^2 \mathbf{G}_{\text{mod}}^\top \mathbf{H}\mathbf{G}_{\text{mod}})/B$$

Applying the optimal learning rate, the optimal per-sample per-iteration improvement, $\max_\eta \Delta L/B$, simplifies to

$$\frac{1}{2}\frac{|\mathbf{G}^\top \mathbf{G}_{\text{mod}}|^2}{B\mathbf{G}_{\text{mod}}^\top \mathbf{H}\mathbf{G}_{\text{mod}} + \text{tr}(\mathbf{H}\mathbf{\Sigma}_{\text{mod}}) + \sigma^2 \text{tr}(\mathbf{H})/(Bc^2)} \tag{16}$$

Again, the decelerator is the same as (8). Therefore, DP training will be as good as the standard training when the decelerator is small under the modified loss. This supports the observation that DP adversarial training can be accurate in [15], if DP natural training is comparable to non-DP natural training.

## A.7 Explaining the effectiveness of DP continual pre-training

Taking a step further than Remark 4.1, we claim that DP continual pre-training can converge as fast as non-DP pre-training (FullyPublic), conditioning on that the non-DP initialization is sufficiently

strong: running $sT$ iterations of public training followed by the DP training is only marginally weaker than FullyPublic in (7):

$$\sum_{t=1}^{sT} \frac{|\mathbf{G}_t|^4}{\mathbf{G}_t^\top \mathbf{H}_t \mathbf{G}_t + \frac{\text{tr}(\mathbf{H}_t \mathbf{\Sigma}_t)}{B}} + \sum_{t=sT}^{T} \frac{|\mathbf{G}_t|^4}{\mathbf{G}_t^\top \mathbf{H}_t \mathbf{G}_t + \frac{\text{tr}(\mathbf{H}_t \mathbf{\Sigma}_t)}{B}} + (8)$$

$$\lesssim \sum_{t=1}^{sT} \frac{|\mathbf{G}_t|^4}{\mathbf{G}_t^\top \mathbf{H}_t \mathbf{G}_t + \frac{\text{tr}(\mathbf{H}_t \mathbf{\Sigma}_t)}{B}} + \sum_{t=sT}^{T} \frac{|\mathbf{G}_t|^4}{\mathbf{G}_t^\top \mathbf{H}_t \mathbf{G}_t + \frac{\text{tr}(\mathbf{H}_t \mathbf{\Sigma}_t)}{B}},$$

given that the decelerator (8) is small for $t > sT$.

# B    Experiment settings and additional details

In this section, we provide additional descriptions and experiment details for the numerical results in the main paper.

## B.1    Datasets

We use the following datasets throughout this paper.

- ImageNet: ImageNet-21k [25] is the full dataset with various releases which contain 14.2M images from 21,841 classes. ImageNet-1k is a subset of ImageNet-21k using 1k high-level classes. We use the ILSVRC2012 version, which contains 1.2 million training images and 100000 test images. ImageNet-11k [70] is another subset of ImageNet-21k that removes invalid/infrequent classes, retaining 11.1M images (train:test=10.5M :0.52M ) and 10,450 classes in the Winter21 version.

- CIFAR-10/CIFAR-100: 50,000 training and 10,000 test images, with 10 or 100 classes, respectively.

- Food101: it contains 101 classes of food, with 101,000 images in total, 1000 images per class (training 750 and test 250).

- SVHN: it contains 10 classes of digits in natural scene images, with 73257 training and 26032 test images.

- Aircraft: FGVCAircraft dataset contains 3334 training and 3333 test images with 100 classes.

- Places365: Places365-Standard dataset contains 1.8 million training and 36000 test images from 365 scene classes.

- iNat2021: iNaturalist 2021 dataset contains 10,000 classes of species, with 2.7 million training and 0.1 million test images.

- CodeParrot: a GitHub dataset of about 180 GB containing roughly 20 million Python files. We use 3% of these files, that is 606,720 rows of training and 3322 rows of test data. We take sequence length 128.

- E2E: a dataset of restaurant reviews, containing 42,061 training and 4693 test instances. We take sequence length 100.

## B.2    Experiment settings

For images, we use $224 \times 224$ resolution and patch 16 for vision transformers. After pre-training, an additional classifier head is inserted for each downstream task.

### B.2.1    Figure 1

We use batch size 1000 and search over learning rate $\in [5e-5, 1e-4, 2e-4, 5e-4]$.

1. CIFAR10, ViT-Base, random initialization, $\eta = 5e-5$ without noise, $\eta = 5e-4$ with noise, 10 epochs.

2. CIFAR100, ViT-Large, pretrained by `timm` library, $\eta = 5e - 5$ without noise, $\eta = 5e - 4$ with noise, 5 epochs.

3. CodeParrot, GPT2-Large, random initialization, $\eta = 1e - 4$ without clipping, $\eta = 5e - 4$ with clipping. This follows closely from `https://huggingface.co/learn/nlp-course/chapter7/6`.

4. E2E, GPT2-Large, pretrained by `transformers` library, $\eta = 1e - 4$ without noise, $\eta = 5e - 4$ with noise, 10 epochs.

### B.2.2 Figure 3

$n = 10^6, \epsilon = 1, \delta = 1/n, S = 10^6$ (1 epoch). We use the privacy accountants in Opacus [88]: RDP [59], GDP [26, 8], PRV [34]. The dashed lines depict $\sigma(1)^2/B$ for these privacy accountants.

### B.2.3 Figure 5

For pre-training, $\sigma^2 = 0.25, \mathbf{G}^\top \mathbf{H} \mathbf{G} = 1e2, \text{tr}(\mathbf{H})/c^2 = 2e8, \text{tr}(\mathbf{H}\boldsymbol{\Sigma}) = 2e4$; for fine-tuning, $\sigma^2 = 0.25, \mathbf{G}^\top \mathbf{H} \mathbf{G} = 1e2, \text{tr}(\mathbf{H})/c^2 = 2e6, \text{tr}(\mathbf{H}\boldsymbol{\Sigma}) = 2e4$.

### B.2.4 Figure 6

We train ViT-Base with $\eta = 5e - 4, B = 500, \epsilon = 2$. We use the Hutchinson method to compute Hessian-related trace, e.g. $\text{tr}(\mathbf{H}) = \mathbb{E}_{\mathbf{v} \in \mathbb{R}^d}[\mathbf{v}^\top \mathbf{H} \mathbf{v}] \leftarrow \frac{1}{k} \sum_{i=1}^{k} \mathbf{v}_i^\top \mathbf{H} \mathbf{v}_i$, where $\mathbf{v} \sim N(0, I_d)$ and $k = 100$. Note the Hessian-vector product $\mathbf{H}\mathbf{v}$ can be computed by one back-propagation of $\mathbf{G}(\mathbf{w})^\top \mathbf{v} \in \mathbb{R}$ on $\mathbf{w}$.

### B.2.5 Figure 7

We train GPT2-small with $\eta = 2 \times 10^{-4}$, $B = 7680$, epochs $E_{\text{pub}} = 1$ (if not early stopped by patience $= 2$), $E_{\text{priv}} = 3$ for mixed training, $E = 4$ for other cases. In the figure, 'PubRatio' is the percentage of public data among all data.

### B.2.6 Table 5

We empirically observe that ViT-Base models with/without fine-tuning on ImageNet1k.

E.g. `vit_base_patch16_224_miil.in21k_ft_in1k` v.s. `vit_base_patch16_224_miil.in21k`;

`vit_base_patch16_224.augreg_in21k_ft_in1k` v.s. `vit_base_patch16_224.augreg_in21k` are very similar in final accuracy, though those with fine-tuning converge faster at early epochs.

We train for 10 epochs, $B = 1000$.

- non-DP fine-tuning: full-parameter, $\eta = 5e - 5$ (except for SVHN and CIFAR10 under $\epsilon = 2$, we use $\eta = 2e - 5$ instead).

- DP fine-tuning: BiTFiT [12], $\eta = 2e - 3$ (except for SVHN and CIFAR10 under $\epsilon = 2$, we use $\eta = 5e - 4$ instead). We observe that for our models, the accuracy may benefit from a smaller learning rate (1e-3) when $\epsilon \geq 8$.

### B.2.7 Table 6

We train with full-parameter fine-tuning and $B = 1000$. Our learning rate $\eta = 5e - 5$ ($\epsilon = 8$) and $\eta = 2e - 5$ ($\epsilon = 2$). $x$-shot means $x$ samples per class, e.g. 30-shot CIFAR100 means 3000 samples in total.

### B.2.8 Table 7

We train with $\eta = 5e - 5$ ($\epsilon = 8$) and $\eta = 2e - 5$ ($\epsilon = 2$) for full-parameter fine-tuning and parameter-efficient fine-tuning. We note that with much heavier (public) pre-training, e.g. on JFT4B, DP NFnet-F3 and JFT4B can achieve even better accuracy [24].

### B.3 Details in training stage switching

When switching from public pre-training to private continual pre-training, the DP gradient may have a different scale than the historical optimizer states and make the continual pre-training unstable. Therefore, we investigate different strategies to re-initialize the optimizer states during the switching. Throughout the training, we use AdamW, which have three states: the iteration index ($t$), the first-order momentum ($\mathbf{m}$), and the second-order momentum ($\mathbf{v}$).

In the experiment, we train a ViT-Base model from scratch on the CIFAR100 dataset. In the first three epochs, we use vanilla AdamW, then we switch to DP-AdamW and continue training for one epoch. During the switching, we fixed the learning rate and reset different states ($t, \mathbf{v}, \mathbf{m}$) to zeros. The ablation study result is shown in Figure 8.

From the figure, we observe that re-initializing the first-order momentum $\mathbf{m}$ (R1) results in the best performance when switching from public to private training. For the other cases, we observe a performance drop when switching.

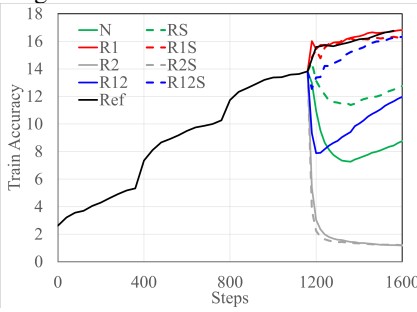

Figure 8: Ablation study of switching from non-DP to DP training with AdamW on CIFAR100 dataset. When switching ($T = 1200$), we re-initialize different states in the AdamW optimizer in different linestyles. "R1", "R2", and "RS" indicate $\mathbf{m}$, $\mathbf{v}$ and $t$ are re-initialized, respectively. "N" indicates no re-initialization, and "Ref" is the reference behavior of continual training with non-DP AdamW.

### B.4 Details for MIA

To conduct the membership inference attack (MIA) in Table 8, we employ a white-box attack with full access to model parameters and data, to evaluate the data protection by our DP pre-training. The attack has two major steps. **Step I: construct MIA dataset.** To construct the MIA dataset, we use the following procedures: 1) for each image in ImageNet-11k, we compute its output logits and loss, which serves as the feature of the MIA dataset; 2) we randomly select $50\%$ of the testing images and the same number of training images ($522, 496$ samples) as the MIA test set, and the rest $50\%$ of the testing images and $10\%$ of the training images as MIA train set; 3) we label the training images as class "1" and testing images as class "0". This creates the MIA dataset with 11k features and binary labels. **Step II: evaluate MIA performance.** After obtaining the MIA dataset, we fit a binary logistic regression with the MIA training set to classify whether an image belongs to the training set of ImageNet-11k (class "1"). We use the L-BFGS optimizer and class re-weighting to train the model for 50 epochs. After training, we evaluate the performance of the training classification model on the MIA test dataset and compute the classification accuracy, precision, recall, F1-score, and AUROC of the classification task. The MIA attack procedures are illustrated in Figure 9.

## C Related works

**Convergence analysis of DP training (the goals)** Recent works have attributed the slow DP convergence to the large number of model parameters [95, 48], the noise level added to gradients [84, 14, 86], and the per-sample gradient clipping [11, 21, 94]. Our analysis in (6) and (8) covers these factors as well as the choice of hyperparameters. We note that a key quantity of DP convergence is $\text{tr}(\mathbf{H})$, which implicitly covers the model dimension $d$ and is also analyzed in [54] to give a dimension-free generalization bound. Moreover, existing works mostly study the empirical convergence on the training set, in terms of the gradient norm $||\mathbf{g}_t|| \to 0$, the parameter space $\mathbf{w}_t \to \mathbf{w}_*$, or the training loss $L_t \to L_*$ as $t \to \infty$. Our work focuses on the *generalization* performance,

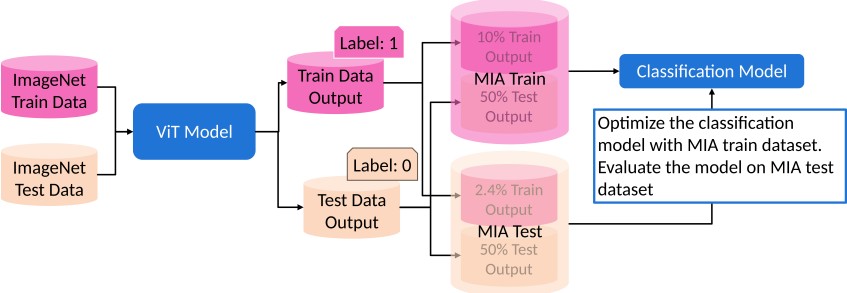

Figure 9: The process of membership inference attack (MIA).

and pays particular attention on the *data efficiency* (as well as computation efficiency) that is rarely captured in the literature.

**More on DP convergence analysis (the assumptions)**   The convergence analysis of non-convex DP deep learning is very challenging and under-studied. There are roughly 3 routes to walk around the challenge: (1) Working on convex models instead (see [73, 23] and more in Sec 4.1.2 [66]), e.g. assuming some form of convexity or only optimizing the last layer (essentially a linear model). This route offers deep theoretical insights thanks to the simplified models, but generally fails to match the training dynamics nor the performance of deep learning. For instance, last-layer training may work reasonably well in computer vision but not so in language tasks, obtaining only 26 BLEU score for DP-GPT2 on E2E dataset compared to 63 BLEU score via DP full-parameter training. (2) Working on the continuous-time gradient flow, rather than the gradient descent in practice, in order to get rid of the per-sample gradient clipping or the Gaussian noise. This route essentially works with infinitely small learning rate, which falls short in the performance as SOTA DP models are trained with large learning rate [49]. (3) Assuming that the loss $L$ is Lipschitz continuous (i.e. the gradient is bounded) [3, 81, 82] or Lipscthiz smooth [21, 14, 86]. While both assumptions lead to some insights of the DP training behaviors, the Lipschitz constant is hardly calculable and time-dependent (maybe even diverging) in practice. Particularly, in the Lipschitz continuous setting, the per-sample gradient clipping is not used in DP deep learning, reducing the technical difficulty significantly but missing the gist of DP-SGD from a practitioner's viewpoint.

In contrast, our work does not rely on these and many other assumptions in the literature, because we do not study the loss convergence along all iterations. We instead scrutinize the per-iteration loss improvement in (5). Put differently, we focus on the local behavior of DP optimizer rather than the global behavior.

**Differentially private training**   The literature of DP deep learning has predominantly focused on fine-tuning the pre-trained models (e.g. full-parameter [1, 49, 14] and PEFT [78, 89, 12, 58]). Importantly, DP fine-tuning (1) can be as performant as the standard non-DP fine-tuning consistently, (2) achieves better empirical performance with larger models [14, 9] and (3) necessarily relies on the public data pre-training. Our work is distinct from previous works since we focus on DP pre-training with large sample size, large number of iterations, and very limited public data. Interestingly, we notice that DP pre-training also enjoy these desirable characteristics of DP fine-tuning, e.g. requiring public data (only a small amount) in Section 4.1 and benefiting from scaling up the models (see Figure 4, Figure 5 and Table 9 in [90]). This empirical evidence of similar training dynamics between DP pre-training and DP fine-tuning, despite their difference in learning goals, is well-supported by our decelerator analysis in Section 3.

**Continual pre-training**   Continual pre-training is a training strategy that accumulates knowledge from a large amount of data, in order to improve its generalization ability to unseen domains and datasets [36, 28, 83, 43, 72]. It is different from fine-tuning which focuses on a task-specific and often smaller dataset. Therefore, the (continually) pre-trained models possess strong few-shot (includign zero-shot) ability but may not be competent in specific functionality such as conversation, and vice versa for fine-tuned models like ChatGPT, Alpaca [76], and Dolly [22].

Multiple DP pre-trained models have been developed [47, 24, 90], but these models either (1) suffer from low accuracy, e.g. 6.9% on ImageNet-1k without additional data by [47] and 32.4% by [24]; or (2) demand significantly more data and compute to match the non-DP pre-training, e.g. $100\times$ in

[90] when comparing DP ViP model to non-DP SimCLR; or (3) rely on uncommon tricks such as a huge batch size from 16-98K, that are not adopted in the standard deep learning community. Related to (3), we believe DP continual pre-training can further improve using the existing techniques from the non-DP continual training, including the replay mechanism and curriculum learning, and avoid catastrophic forgetting.

**Hessian-based analysis in deep learning**    Applying the second-order Taylor expansion of loss motivates the famous Newton's method and is commonly adopted in deep learning [57, 97, 85, 56, 91], where the Hessian matrix is useful to analyzing the convergence and data efficiency (e.g. selecting the critical batch size [57]), even though it is infeasible to derive $\mathbf{H} \in \mathbb{R}^{d \times d}$ explicitly for large models. Our work follows the same path with a specific focus on DP related operations (i.e. the clipping and the noise). We use the Hutchinson method and Hessian-vector product to compute $\text{tr}(\mathbf{H})$, $\mathbf{HG}$ and so on.

**System design of DP training**    To make DP deep learning broadly applicable, we believe it is necessary to not only evaluate DP algorithms on the utility, but also from a system design perspective. The design of a DP system, such as our DP continual pre-training, should resemble that of the standard non-DP system (see our extensive discussion in Remark 4.2). Such a design will be compatible to and benefit from new advances in the much larger non-DP literature, unifying DP and non-DP communities, instead of crafting techniques that are limited to DP learning only.

**Data distribution shift**    There may be some distribution shift between the *public* pre-training data and the *private* fine-tuning data or continual pre-training data. Empirically speaking, DP training can be robust to such distribution shift. For instance, DP fine-tuning has successfully transferred from ImageNet to CIFAR10, CIFAR100, SVHN, Food101, CelebA, FMNIST, GSTRB [14], from JFT to ImageNet [24, 58], and from Places365 to ImageNet [47]. Although this work does not address the distribution shift due to similarity between ImageNet-1k and ImageNet-11k, our analysis in Section 3, especially the formula of decelerator, still holds in this scenario. Therefore, we expect our DP continual pre-trainin to work withstanding the data distribution shift, as empirically demonstrated from Shaders to LAION [90].

**Concurrent work**    The concurrent work – ViP [90] – also studies the DP continual pre-training, thus is the closest strategy to ours. We elaborate the similarity and differences between two strategies.

Similarity

1. Both use ViT-Base as backbone and AdamW as optimizer.
2. Both use self-supervised learning (SSL) for public pre-training.
3. Both use $\epsilon = 8$ for DP continual pre-training.
4. Both public datasets (Shaders v.s. ImageNet-1k) have $\approx$ 1M images.

Differences

1. ViP uses SSL on DP continual pre-training. We use supervised learning.
2. ViP adds a decoder to ViT-Base (in total 99.0M parameters) during the pre-training. We replaces the classifier head of ViT-Base (94.4M parameters) for DP continual pre-training. Hence the model architecture is different.
3. ViP trains on Shaders (for 1.6B images) and LAION (for 0.6B images). We train on ImageNet-1k (for 0.3B images) and ImageNet-11k (for 0.4B images). Hence data distribution and quality is different, and our training requires about 1/3 the computation.
4. ViP's private dataset (LAION) has 233M images. Ours (ImageNet-11k) has 12M images. Hence our training requires much smaller dataset (see Figure 3(a) [90]).
5. ViP has to use huge batch size 98K (see Figure 4(b) [90], where the loss diverges with batch size 8K). We use 4K.
6. ViP experiments with various model sizes, from ViT-Nano (18.6M) to Large (233M). We only use ViT-Base.

# D   Algorithm of DP continual pre-training

**Algorithm 1** DP continual pre-training

1: switch_to_DP=False
2: **for** $t = 1, 2, ...$ **do**
3:     Compute the loss $L_t$ by forward pass
4:     **if** switch_to_DP==False: **then**
5:         Compute public gradient **g**
6:     **else**
7:         Compute private gradient **g**
8:     **end if**
9:     Update $\mathbf{w}_{t+1} = \mathbf{w}_t - \eta \mathbf{g}$
10:    **if** $L_t > L_{t-1}$ **then**
11:        Set switch_to_DP=True
12:    **end if**
13: **end for**

