# OpenReview forum: "Pre-training Differentially Private Models with Limited Public Data"
_NeurIPS.cc/2024/Conference — NeurIPS 2024 poster_

### Official Review · Reviewer_wr8p · 2024-06-15

**Soundness:** 3
**Presentation:** 2
**Contribution:** 2
**Rating:** 4
**Confidence:** 4

**Summary:**

This work theoretically justifies the loss of utility under DP pre-training under the lens of a Hessian. This theoretical result can then be leveraged to perform efficient pre-training of models on public data (small amount) to significantly improve DP-trained model's performance.

**Strengths:**

The problem authors attempt to solve in this paper is very important and the work is timely (given the concerns re:usage of copyrighted material during pre-training of large-scale models). To the best of my knowledge, the results are theoretically justified and sound. There are many experiments supporting authors' claims in this work, outperforming the existing approaches.

**Weaknesses:**

My fundamental issue with the positioning of this work is that we already know that public data improves DP - [1,2,3,4,6]. We also know that DP training during pre-training is harming utility more than it would have harmed the pre-trained model [2,3,4]. Therefore my question is what is the main scientific output of this paper? The conclusions are already widely established and I do not see any new outcomes that can be derived from the conclusions of this work. It would have been interesting to see if this result it linked to model's generalization/memorization capacities as described in [5], showing some novel contextualized understanding of the training dynamics, but there was no discussion on the interplay of the training time and memorization capacity under DP.

The theoretical justification of why this phenomenon arises is indeed interesting and can be useful, but on its own does not provide any novel insights into the training process that are 'actionable'. This is, again, because it was already previously established that a) DP pre-training harms utility [1,6,7], b) public data aids utility of DP models [2] and c) when models are untrained, they are more susceptible to various phenomena that can affect the training dynamics (e.g. loss of information via clipping [5]). So I am struggling to understand what can one gain from the results of this work? The 10% data threshold is a purely empirical result, which may or may not scale to other settings and datasets.

[1] - Bu, Zhiqi, Jialin Mao, and Shiyun Xu. "Scalable and efficient training of large convolutional neural networks with differential privacy." Advances in Neural Information Processing Systems 35 (2022): 38305-38318.
[2] - Kurakin, Alexey, et al. "Toward training at imagenet scale with differential privacy." arXiv preprint arXiv:2201.12328 (2022).
[3] - Nasr, Milad, et al. "Effectively using public data in privacy preserving machine learning." International Conference on Machine Learning. PMLR, 2023.
[4] - Ganesh, Arun, et al. "Why is public pretraining necessary for private model training?." International Conference on Machine Learning. PMLR, 2023.
[5] - Feldman, Vitaly. "Does learning require memorization? a short tale about a long tail." Proceedings of the 52nd Annual ACM SIGACT Symposium on Theory of Computing. 2020.
[6] - Mehta, Harsh, et al. "Large scale transfer learning for differentially private image classification." arXiv preprint arXiv:2205.02973 (2022).
[7] - Mireshghallah, Fatemehsadat, et al. "Differentially private model compression." Advances in Neural Information Processing Systems 35 (2022): 29468-29483.

**Questions:**

Can the framework be used to explain the reason behind DP being 'more difficult' for untrained models through the prism of memorization?

Is there a more theoretically sound way to define the threshold of public data that is required for appropriate performance?

**Limitations:**

As outlined above: while the theoretical justification of the results is indeed interesting, there are no novel conclusions our takeaways from this work that we have not previously had from similar works in the area of DP training.

---

> ### Author Rebuttal · Authors · 2024-08-06
>
> We thank the reviewer for the insightful comments. We will address them point-to-point and hope the reviewer can raise the score if satisfied.
>
> > My fundamental issue with the positioning of this work is that we already know that public data improves DP - [1,2,3,4,6]. We also know that DP training during pre-training is harming utility more than it would have harmed the pre-trained model [2,3,4]. Therefore my question is what is the main scientific output of this paper? The conclusions are already widely established and I do not see any new outcomes that can be derived from the conclusions of this work.
>
> We would like to highlight that the main scientific output is to answer "why DP training suffers from slow convergence?", whereas existing works mostly only observe that "DP convergence is slower than non-DP" without a fine-grained explanation. That is, knowing something happens is not the same as knowing why something happens, and scientific insights beyond empirical can only roots from the why. Our explanation via the trace of Hessian works for both pre-training and fine-tuning (our contribution 1; see Figure 1). We specifically separate the clipping and noising and emphasize that clipping is not troublesome but noising is (our contribution 2; again see Figure 1)! We then use our analysis to provide an actionable training strategy that only requires a small amount of pre-training data (which is very different from most works where most data are non-DP and only a fraction of finetuning data requires DP) and automatically switches to DP training (our contribution 3).
>
> > The theoretical justification of why this phenomenon arises is indeed interesting and can be useful, but on its own does not provide any novel insights into the training process that are 'actionable'. This is, again, because it was already previously established that a) DP pre-training harms utility [1,6,7], b) public data aids utility of DP models [2] and c) when models are untrained, they are more susceptible to various phenomena that can affect the training dynamics (e.g. loss of information via clipping [5]). So I am struggling to understand what can one gain from the results of this work?
>
> We are glad the reviewer finds our theoretical justification interesting. We would like to emphasize some actionable insights in our work: (1) The analysis in Section 3 shows that noising is the main cause of slow convergence. Therefore, we recommend a future direction to improve DP training by reducing the noise vector, instead of improving on the per-sample clipping. Some efforts may include parameter-efficient training (like LoRA), pruning and noise reduction (e.g. via tighter privacy accounting in Implication 2.3). (2) The Section 4 and all experiments rely on our continual pre-training strategy, where we monitor the loss to switch from non-DP to DP training (which implicitly uses the tr(H) information). (3) Given that tr(H) is worsening DP convergence, we may adopt sharpness-aware minimization and regularization to encourage the model to move in a flat region, so that DP convergence can accelerate.
>
>
> > Can the framework be used to explain the reason behind DP being 'more difficult' for untrained models through the prism of memorization?
>
> We focus on the training dynamics, which the memorization may not directly link to.
>
> > Is there a more theoretically sound way to define the threshold of public data that is required for appropriate performance?
>
> We show DP convergence is always going to be slower than non-DP, despite the slowdown can be insignificant after initial training, so the threshold will be subjective to "what level of slowdown is acceptable?". Generally speaking, if DP batch size is chosen properly (see Line 184), we should expect at most 2 times slowdown, and more non-DP pre-training brings down this factor of 2 to close to 1.

---

> > ### Comment · Reviewer_wr8p · 2024-08-09
> > **Response to the rebuttal**
> >
> > I would like to thank the authors for their responses! I still have some points I would like to note.
> >
> > From the discussion with reviewer JCrd I am not sure I agree with your response to them:
> >
> > >  the extent of the bias is less significant and hence ignorable, compared to the effect of DP noising
> >
> > This may very well be the case for the specific settings of your experiments, but I am not convinced this is a general 'rule' you can follow and claim that this bias is 'ignorable'.
> >
> > >  and noise reduction (e.g. via tighter privacy accounting in Implication 2.3)
> >
> > When I said 'actionable results', telling the community to use better DP tools (i.e. tighter accountant) is not really something actionable from YOUR results (i.e. we would always use the tightest accounting all else aside anyway).
> >
> > After reading the rest of the response, I am keeping my score unchanged.

---

> ### Author Response · Authors · 2024-08-09
>
> Thank you for joining the discussion! We understand your concern that the clipping bias may be non-ignorable in some settings that we haven't covered. From our experience with DP model training in all projects that consist of hundreds of experiments, we always observe that "clipping without noising" gives similar performance to non-DP empirically. Nevertheless, we can not rigorously claim this regardless of how many settings/experiments we test. Hence we would state "this analysis only approximates the scenario where the clipping introduces ignorable bias, e.g. when the clipping threshold is large or when unbiased clipping is used (c.f. 'Differentially Private SGD Without Clipping Bias: An Error-Feedback Approach')".
>
> Additionally, we highlight that our connection between tr(H) and noising (to which we attribute the slowdown of DP) still holds even if we take the clipping bias into consideration: Equation (5) will become
> $$\Delta L_\text{priv}=\eta\hat{G}^\top G-\frac{\eta^2}{2}\left(\hat{G}^\top H \hat{G}+\frac{tr(H\hat{\Sigma})}{B}+\frac{\sigma^2 tr(H)}{B^2}\right)$$
> where the hat stands for **biased gradient** from per-sample clipping.
>
> Regarding the actionable items specifically from our results, we would recommend noise reduction like low-pass filtering (we are working on it) and sharpness-aware minimization (to reduce tr(H), known as the sharpness). It would also be desirable to experiment our DP continual training on more settings such as NLP. Happy to extend the discussion!
>
> Given that this work is on the borderline, we would appreciate it if the reviewer can consider raise the score.

---

### Official Review · Reviewer_9edf · 2024-06-25

**Soundness:** 3
**Presentation:** 3
**Contribution:** 3
**Rating:** 5
**Confidence:** 4

**Summary:**

This paper addresses the challenge of DP pretraining, which has been limited due to performance degradation. The authors first analyze per-iteration improvements of the optimization process and then propose a novel DP continual pre-training strategy that uses a small amount of public data to mitigate the negative effects of DP noise. They then empirically demonstrate the effectiveness of their approach on large-scale vision models.

**Strengths:**

- Conclusions are well supported by figures and illustrations.
- The framework of analyzing improvements appears novel to me.
- The experimental evaluations are extensive, and the improvements are significant.

**Weaknesses:**

- The authors claim that *a certain amount of public data can mitigate deceleration*. However, Section 4 lacks discussion about deceleration. The given claim is that per-iteration improvement is increased, which is obvious given public data. Also, it's unclear why *limited* public data is emphasized, as the analysis does not involve public sample sizes. This makes the article somewhat incoherent, as the interesting theoretical analysis in Section 3 does not provide strong suggestions for the methodology.
- The experiments lack some ablation studies to support the key methodological proposal.
- Citations should be updated to published versions rather than arXiv preprints where possible.
- In line 14, there's an extra "and".

**Questions:**

Overall, this paper is interesting. I am happy to raise my score if some of the questions are addressed.


- Can the authors provide some comments on the first weakness?
- Section 3.3 is somewhat confusing. First, Remark 3.1 investigates the differences between pre-training and fine-tuning by comparing $B_{non-DP} G^{\top} H G$ and $tr(H\Sigma)$, while in fact, public per-iteration improvement is monotonic with respect to $B$. Why use a comparison between $B_{DP}$ and $B_{non-DP}$ for deriving the explanation? Also, can the authors more rigorously explain what is meant by "data-efficient"?
- Moreover, the difference between public improvement (7) and private improvement (6) lies in the decelerator $\sigma^2 tr(H) / (Bc^2)$, which seems smaller for pre-training where the loss landscape is flatter and thus the curvature $tr(H)$ is smaller. What is wrong with this intuition?
- It is observed that prediction accuracy is linear in $\log B$, for instance in Figure 4 in [1], while the statement here is that $B$ should be moderately chosen. Though these are not the same quantity, could the authors comment on the impact of $B$ on DP-SGD?
- Based on analysis in Section 3, is there any off-the-shelf rule for choosing $B$?
- Can the authors provide some ablation studies (can be on toy datasets) on the impact of $s$?



[1] Tom Sander, Pierre Stock, and Alexandre Sablayrolles. Tan without a burn: Scaling laws of dp-sgd. In International Conference on Machine Learning, pages 29937–29949. PMLR, 2023.

**Limitations:**

The limitations should appear in the main text.

---

> ### Author Rebuttal · Authors · 2024-08-06
>
> We thank the reviewer for the time and the comments. We will address them point-to-point and hope the reviewer can raise the score if satisfied.
>
> > The authors claim that a certain amount of public data can mitigate deceleration. However, Section 4 lacks discussion about deceleration. The given claim is that per-iteration improvement is increased, which is obvious given public data. Also, it's unclear why limited public data is emphasized, as the analysis does not involve public sample sizes. This makes the article somewhat incoherent, as the interesting theoretical analysis in Section 3 does not provide strong suggestions for the methodology.
>
> Indeed, we agree with the reviewer that we do not provide a theoretical characterization of "limited public data", whereas the "limited" is only emphasized from a practical standpoint. In Section 4, we have leveraged our knowledge about the deceleration in Equation (9) to suggest the mixing ratio of public and private gradient, and to motivate the DP continual pretraining that empirically mitigates the deceleration by achieving higher utility. We emphasize the "limited public data" because our mitigation of deceleration is so effective that we use much fewer public data than other DP training strategies (see right-most column in Table 2).
>
> > The experiments lack some ablation studies to support the key methodological proposal.
>
> We have ablation studies over privacy budgets (multiple epsilon), few-shot or full settings, task difficulties (from CIFAR10 to Places365). We are happy to discuss more ablation studies if the reviewer can be more specific.
>
> > Citations should be updated to published versions rather than arXiv preprints where possible.
>
> We complete agree and will update in the camera-ready (NeurIPS does not allow revision this year).
>
> > Section 3.3 is somewhat confusing. First, Remark 3.1 investigates the differences between pre-training and fine-tuning ..., while in fact, public per-iteration improvement is monotonic with respect to B. Why use a comparison ... for deriving the explanation? Also, can the authors more rigorously explain what is meant by "data-efficient"?
>
> Here "data-efficient" means the per-sample improvement is high: the same 0.1 improvement obtained by 10 data samples is much more data-efficient than obtained by 1000 samples. While public per-iteration improvement is monotonic, the rate of improvement can be different. For example, if we have the (batchsize, loss improvement) pairs: (B=1, 0.1), (B=2, 0.08), (B=10, 0.07), (B=1000, 0.0699). Hence, in Remark 3.1, we stated that non-DP batch size should be small to be data-efficient.
>
> > Moreover, the difference between public improvement (7) and private improvement (6) lies in the decelerator, which seems smaller for pre-training where the loss landscape is flatter and thus the curvature is smaller. What is wrong with this intuition?
>
> We are glad the reviewer points this out! We agree that tr(H) is small initially as we also demonstrate in Figure 6. But it quickly increases to a large value (within 5 epochs) for a long time (say epoch 5 to 40) before it decreases again. Therefore, even though DP training is fast initially, it is only fast for a short period and overall DP is much slower than non-DP. This is also observed in Figure 1 (a)(c). We will add this discussion to the camera-ready paper.
>
> > It is observed that prediction accuracy is linear in log(B), for instance in Figure 4 in [1], while the statement here is that B should be moderately chosen. Though these are not the same quantity, could the authors comment on the impact of B on DP-SGD?
>
> We believe B should be moderately chosen. However, the optimal B may be very large and hence, for a range of batch sizes smaller than the optimal B, it is not wrong empirically that increasing B could improve convergence. Nevertheless, we notice [1] does not explicitly consider adjusting the learning rate for different batch sizes. Therefore, the conclusion there does not hold in our setting as we also use optimal learning rate (see Equation (5)) in this work. Notice that in Equation (5), if eta learning rate is independent of B, then it is monotonically decreasing in B. However, in theory, and in practice, the learning rate depends on B, and thus Equation (5) is NOT monotonically decreasing in B, as we demonstrate in Figure 5.
>
> > Based on analysis in Section 3, is there any off-the-shelf rule for choosing B?
>
> Unfortunately, choosing B is not easy, even for non-DP training.
>
> > Can the authors provide some ablation studies (can be on toy datasets) on the impact of s?
>
> In Figure 7, we have three values of s: s=0 is fully private training (red curve); s=1 is fully non-private training (black curve); automatic s (switch point and blue curves) around 25\%. We also use automatic s (around 10\%) in all experiments in Section 5. In short, if we plot utility against s, we will observe a sharp trade-off that small s (not too small) suffices to pick up most utility.

---

> > ### Comment · Reviewer_9edf · 2024-08-09
> >
> > Thanks for the detailed clarification and I have updated my score.

---

### Official Review · Reviewer_JCrd · 2024-07-08

**Soundness:** 3
**Presentation:** 3
**Contribution:** 3
**Rating:** 7
**Confidence:** 4

**Summary:**

This paper proposed a Hessian-based analysis of the per-iteration loss for DP-SGD and non-private training, and provide a theoretical explanation for slower convergence of differentially private training. Authors identified the *decelerator* compontent in per-iteraton loss improvement, associated with the gradient clipping and noising, providing a theoretical backing for differences between DP impact on pre-training vs. fine-tuning. Using the same framework, authors also suggest the derivation for the optimal batch size for private training under the fixed computation budget (i.e. fixed number of samples processed), which strikes the balance between lower per-sample noise and training slowdown associated with growing batch size.

Based on this analysis above, authors suggest the impact of DP on convergence can be mitigated by public pre-training data. While using public pre-training data to complement DP fine-tuning has been explored before, authors emphasize the difference between fine-tuning and continual pre-training, provide a theoretical analysis for the optimal mixing proportion for private/public data, and evaluate their models after private continual pre-training on a downstream fine-tuning tasks.

**Strengths:**

I believe this is a strong and well-written paper with valuable contributions to the field.

* The paper is very clearly written, with all major claims supported by arguments, derivations, and graphs. It's easy to follow and it gets the point across.
* The derivation of the decelerator component of the per-iteration loss is a valuable contribution. It provides a good explanation for the DP-SGD training dynamics, and provides a theoretical backing for the previously observed empirical results. Specifically Fig. 6 provides very interesting insights intro DP-SGD training which I haven't seen analyzed before.
* The downstream findings (e.g. optimal batch size or optimal ratio of private/non-private data) could be impactful for the practical applications of DP-SGD.

**Weaknesses:**

The major weakness of the paper (at least its theoretical part) is the assumption that per-sample gradient clipping does not introduce bias into the gradient approximation (line 116). This is a very strong assumption, and something that, to the best of my knowledge, is not generally accepted in the field (e.g. see [this paper](https://proceedings.neurips.cc/paper_files/paper/2020/file/9ecff5455677b38d19f49ce658ef0608-Paper.pdf)). To justify the assumption authors point to Fig. 1, which to me doesn't look like it supports their claim - there's a significant difference between vanilla optimizer and optimizer + clipping in pre-training.

Additionally, the analysis in this paper assumes the oracle knowledge of matrices G (gradient expectaition) and H (Hessian). For practical applications where this is infeasible, it would be useful to look at, e.g. optimal batch size not only from an optimal loss perspective, but also take into account how well the batch gradient approximates the actual gradient G.

Oh the evaluation side, I see a slight disconnect with the theoretical results. For instance, after the results in Sec. 4.1 it would be natural to explore different mixing ratios of public/private data in training - however authors only focus on a setup with fixed pre-training and continual training datasets.

The results presented in Tables 3,4 and 5 do make a good case that the proposed approach is valid, but lack proper baselines. Comparisons are made either with non-private models, models trained on different dataset, or model trained for different number of epochs - contradicting the scenario with fixed compute budget.

Authors do not report compute resources used for the experiments, which is especially relevant for reproducibility of the paper's results, as it works with Hessian matrices which can be very computationally expensive to compute.

**Questions:**

* I think the formatting for Fig.5 is wrong - in the text you refer to "upper left" plot, while in the submitted version it's rendered as a single line of 4 plots
* I would be interested in reading how did you compute the data for Fig.6 - did you explicitly compute full matrices `G` and `H` or used some approximations?
* I don't fully understand why in Fig.5 (pre-training) the blue dashed line is linear, suggesting that $G^THG$ is constant.
* In Table 2, a) does number of images include continual pre-training? and b) what does "non-privacy" column refer to?
* What is the criteria for distinguishing continual pre-training from fine-tuning? For example, in your experiments you perform continual pre-training with a different objective than earlier pre-training (supervised vs non-supervised). Does it not justify the "fine-tuning" term?

**Limitations:**

Authors do not discuss limitations explicitly, but are very upfront about assumptions they make (e.g. fixed compute budget, assuming no bias from clipping, etc). Authors also briefly mention limitations in the checklist (but not the main body of the paper).

---

> ### Author Rebuttal · Authors · 2024-08-06
>
> We thank the reviewer for the positive feedback and comments. We will address them point-to-point. The answers to some questions are merged in the response to the weaknesses.
>
> > The major weakness of the paper (at least its theoretical part) is the assumption that per-sample gradient clipping does not introduce bias into the gradient approximation (line 116). This is a very strong assumption, and something that, to the best of my knowledge, is not generally accepted in the field (e.g. see this paper). To justify the assumption authors point to Fig. 1, which to me doesn't look like it supports their claim - there's a significant difference between vanilla optimizer and optimizer + clipping in pre-training.
>
> We agree that gradient clipping definitely introduces bias to the gradient approximation, but the extent of the bias is less significant and hence ignorable, compared to the effect of DP noising (see the closeness of blue and black curves, and the remarkable distance between yellow and black curves, especially for GPT2). We are using this assumption to prioritize the analysis of noising (which leads to trace of Hessian and decelerator), which is necessary to simplify our analysis.
>
> > Additionally, the analysis in this paper assumes the oracle knowledge of matrices G (gradient expectaition) and H (Hessian). For practical applications where this is infeasible, it would be useful to look at, e.g. optimal batch size not only from an optimal loss perspective, but also take into account how well the batch gradient approximates the actual gradient G.
>
> We agree that oracle G and H is not available (as we marked in footnote 2) and we will extend the discussion on the selection of batch size.
>
> > On the evaluation side, I see a slight disconnect with the theoretical results. For instance, after the results in Sec. 4.1 it would be natural to explore different mixing ratios of public/private data in training - however authors only focus on a setup with fixed pre-training and continual training datasets.
>
> Thanks for pointing out this precious suggestion. To clarify, the ratio is about the re-weighting of gradients. The reason we didn't experiment with other mixing ratios is in Remark 4.2, as these methods are hard to implement and not scalable, and most do not have open-source code.
>
> > The results presented in Tables 3,4 and 5 do make a good case that the proposed approach is valid, but lack proper baselines. Comparisons are made either with non-private models, models trained on different dataset, or model trained for different number of epochs - contradicting the scenario with fixed compute budget.
>
> We agree the comparisons are not perfectly comprehensive and it is computationally expensive to explore them. E.g. VIP is trained on Shaders21k (hence we will double our computation budget at least) and NFnet is trained on JFT (which is not publicly available). We hope the reviewer would agree that this is acceptable as we follow the same experiment setup as in VIP, and the utility of our method is clear even though we use much less compute.
>
> > Authors do not report compute resources used for the experiments, which is especially relevant for reproducibility of the paper's results, as it works with Hessian matrices which can be very computationally expensive to compute.
>
> We are using 1 A100 GPU and we will release the trained models for reproducibility. We actually never computed the Hessian matrix because we only need tr(H), which can be computed via Hutchinson method (briefly mentioned in Line 820): we sample 50 random vectors $z$ and compute $zHz$ as a scalar and then average over 50. Here $zHz$ is computed from finite difference of losses. Please let us know if more details are desired.
>
> > I don't fully understand why in Fig.5 (pre-training) the blue dashed line is linear, suggesting that GHG is constant.
>
> In Figure 5 the illustration is for one iteration so GHG is a constant and B is the variable. GHG does change over iterations as we show in Figure 6.
>
> > In Table 2, a) does number of images include continual pre-training? and b) what does "non-privacy" column refer to?
>
> a) Yes. We stated in caption that it is "the total number of images". b) DP is defined specifically on dataset. If we train on data A non-privately and then on data B privately, then only B has privacy guarantee and A is indicated in "non-privacy". We will clarify this in the camera-ready.
>
> > What is the criteria for distinguishing continual pre-training from fine-tuning? For example, in your experiments you perform continual pre-training with a different objective than earlier pre-training (supervised vs non-supervised). Does it not justify the "fine-tuning" term?
>
> In this work, we consider fine-tuning to be a) on a specific (and usually much smaller) dataset so that some techniques like parameter efficient finetuning (e.g. LoRA) is applicable because the change in parameters are small. However continual pre-training is on a large amount of data for a series of downstream tasks and LoRA won't work. b) the second dataset has a distribution shift, e.g. many DP papers publicly pretrain on ImageNet and finetune on CIFAR10. However, the continual pre-training mostly uses a similar dataset, e.g. we pretrain on 10\% of ImageNet and continue the training on the other 90\%.

---

> > ### Comment · Reviewer_JCrd · 2024-08-13
> >
> > Thanks for the detailed rebuttal.
> >
> > > We agree that gradient clipping definitely introduces bias to the gradient approximation, but the extent of the bias is less significant and hence ignorable, compared to the effect of DP noising <...> We are using this assumption to prioritize the analysis of noising (which leads to trace of Hessian and decelerator), which is necessary to simplify our analysis.
> >
> > I understand the author's point to prioritize the noise over clipping in their analysis. I agree it is justifiable and provide a foundation for the important theoretical analysis presented in the paper. I would not, however, call the clipping bias "ignorable" - I would argue that it remains an important direction for future research to understand the DP training dynamics.
> >
> > I also appreciate clarifications provided, they were useful for me to understand the paper better.
> >
> > Overall, I believe it's a strong paper and choose to maintain my high score.

---

> > > ### Author Response · Authors · 2024-08-13
> > >
> > > We agree "ignorable" is over-stating and that clipping bias is indeed important to DP training. We will surely add the clarification in next revision.

---

### Official Review · Reviewer_KeLK · 2024-07-09

**Soundness:** 2
**Presentation:** 2
**Contribution:** 3
**Rating:** 7
**Confidence:** 3

**Summary:**

This paper provides a theoretical framework to analyze impact of various aspects (parameters) of DP training on the performance of resulting models. The framework uses hessian of per-sample gradient to compute per-iteration loss improvement. Using the framework, the paper shows how DP impacts performance of model in pre-training more than during fine-tuning, and suggests using public data pre-training as a remedy. Finally, based on the observations, the paper proposes a DP continual learning and shows how it can help improve the performance of upstream and downstream tasks.

**Strengths:**

- Approach of the proposed framework can be useful to analyze many DP settings.
- Interesting conclusions especially Implication 2.4 about batch size
- Proposed continual learning approach seems practically useful in privacy sensitive settings
- Few shot accuracy of the proposed approach is impressive

**Weaknesses:**

- Some assumptions, especially about clip norm, need more clarification
- Some of the theoretical claims could be paired with empirical evidence
- Proposed continual pre-training approach needs better explanation

**Questions:**

- It looks like the conclusions made in the paper rely heavily on the assumption about clip norm multiplier being the same for all per-sample grads. The assumption is because in Figure 1 performances of models with and without clipping are similar. But, why would this be always true? If clipping is very aggressive, it will introduce bias and performance will reduce much more than what we see in Figure 1. Maybe it is good to clarify the scope of the assumption.

- In the “Per-iteration improvement of DP-SGD” it says that parameter updates are generally small. When is this true in practice?

- In Table 3, why do performances of many fine-tuning cases reduce from DINO to Ours(eps=2)? I think the performance should always improve? At the same time, few-shot performance of proposed method is significantly higher than DINO in all cases. Can you clarify this difference?

**Limitations:**

- Assumptions made in the work to set up the theoretical framework need better clarification in terms of the scope of theory.
- It would be good to formally write an algorithm of DP continual pre-training, as I found section 4.2 a bit confusing.

---

> ### Author Rebuttal · Authors · 2024-08-06
>
> We thank the reviewer for the positive feedback and comments. Given that the weaknesses are mostly about clarification and explanation, we are happy to address them if the reviewer can be slightly more specific, besides the questions which we address below.
>
>
> > It looks like the conclusions made in the paper rely heavily on the assumption about clip norm multiplier being the same for all per-sample grads. The assumption is because in Figure 1 performances of models with and without clipping are similar. But, why would this be always true? If clipping is very aggressive, it will introduce bias and performance will reduce much more than what we see in Figure 1. Maybe it is good to clarify the scope of the assumption.
>
> We clarify that the clipping norm multiplier is NOT the same for all per-sample grads and this would NOT be always true. We explicitly state in Line 115-116 that "This approximation **only holds when** the directions of vectors ... are very close, i.e., there is **little** per-sample clipping bias." Note little bias must be distinguished from no bias. In fact, we are using the most aggressive clipping, i.e. the normalization which means clipping threshold is infinitely close to 0, but Figure 1 shows the impact of clipping is much less significant than the impact of noising. In short, clipping does has a bias, but this bias is sufficiently small so that the approximation is sufficiently accurate and informative.
>
> > In the “Per-iteration improvement of DP-SGD” it says that parameter updates are generally small. When is this true in practice?
>
> We thank the reviewer for this question. Here are a few cases where the parameter updates are generally small in deep learning: (1) the learning rate is small because the update is a multiplication of learning rate and gradient; or (2) the model size is large so the training experiences a phenomenon known as lazy training (especially in the neural tangent kernel regime); or (3) weight decay is applied so the parameters are within a ball around the initialization and the updates are bounded.
>
> > In Table 3, why do performances of many fine-tuning cases reduce from DINO to Ours(eps=2)? I think the performance should always improve? At the same time, few-shot performance of proposed method is significantly higher than DINO in all cases. Can you clarify this difference?
>
> We confirm that on the pretraining dataset ImageNet, the performance indeed always improve with DP continual training, including Ours(eps=2). Therefore the inconsistency may arise from the dataset distribution shift, i.e. some datasets are less similar to ImageNet so the improvement does not transfer well. We will add this discussion as a future direction!
>
> > It would be good to formally write an algorithm of DP continual pre-training, as I found section 4.2 a bit confusing.
>
> We haved added it in the rebuttal Appendix D.

---

> > ### Comment · Reviewer_KeLK · 2024-08-12
> > **Thanks for the response!**
> >
> > Thanks for the response. Rebuttal clarifies my questions and concerns, and i have raised my score accordingly.

---

### Author Rebuttal · Authors · 2024-08-06

We thank all the reviewers for the comments and put every effort to address them. Please let us know if there are further questions (though revision is not allowed this year). We provide the algorithm of DP continual pretraining in PDF here (Appendix D).

---

### Decision · Program_Chairs · 2024-09-25

**Decision:**

Accept (poster)

**Comment:**

While most of the reviewers were positive about the paper, there were a few concerns regarding the importance of the result, as pre-training DP models with public data is a well-studied problem. There were also concerns regarding some of the theoretical claims and the practical implementation of the DP continual pre-training approach.

We would recommend the authors to include rebuttal discussion (mainly with reviewer wr8p) in a future version of the paper.